



Geoscientific
Model Development

# FVM 1.0: a nonhydrostatic finite-volume dynamical core for the IFS

**Christian Kühnlein**[1], **Willem Deconinck**[1], **Rupert Klein**[2], **Sylvie Malardel**[1,3], **Zbigniew P. Piotrowski**[4],
**Piotr K. Smolarkiewicz**[1], **Joanna Szmelter**[5], **and Nils P. Wedi**[1]

[1]European Centre for Medium-Range Weather Forecasts, Reading, UK
[2]FB Mathematik und Informatik, Freie Universität Berlin, Berlin, Germany
[3]Laboratoire de l'Atmosphére et des Cyclones, Météo-France, La Reunion, France
[4]Institute of Meteorology and Water Management – National Research Institute, Warsaw, Poland
[5]Loughborough University, Leicestershire, LE11 3TU, UK

**Correspondence:** Christian Kühnlein (christian.kuehnlein@ecmwf.int)

**Abstract.** We present a nonhydrostatic finite-volume global atmospheric model formulation for numerical weather prediction with the Integrated Forecasting System (IFS) at ECMWF and compare it to the established operational spectral-transform formulation. The novel Finite-Volume Module of the IFS (henceforth IFS-FVM) integrates the fully compressible equations using semi-implicit time stepping and non-oscillatory forward-in-time (NFT) Eulerian advection, whereas the spectral-transform IFS solves the hydrostatic primitive equations (optionally the fully compressible equations) using a semi-implicit semi-Lagrangian scheme. The IFS-FVM complements the spectral-transform counterpart by means of the finite-volume discretization with a local low-volume communication footprint, fully conservative and monotone advective transport, all-scale deep-atmosphere fully compressible equations in a generalized height-based vertical coordinate, and flexible horizontal meshes. Nevertheless, both the finite-volume and spectral-transform formulations can share the same quasi-uniform horizontal grid with co-located arrangement of variables, geospherical longitude–latitude coordinates, and physics parameterizations, thereby facilitating their comparison, coexistence, and combination in the IFS.

We highlight the advanced semi-implicit NFT finite-volume integration of the fully compressible equations of IFS-FVM considering comprehensive moist-precipitating dynamics with coupling to the IFS cloud parameterization by means of a generic interface. These developments – including a new horizontal–vertical split NFT MPDATA advective transport scheme, variable time stepping, effective precondi-

tioning of the elliptic Helmholtz solver in the semi-implicit scheme, and a computationally efficient implementation of the median-dual finite-volume approach – provide a basis for the efficacy of IFS-FVM and its application in global numerical weather prediction. Here, numerical experiments focus on relevant dry and moist-precipitating baroclinic instability at various resolutions. We show that the presented semi-implicit NFT finite-volume integration scheme on co-located meshes of IFS-FVM can provide highly competitive solution quality and computational performance to the proven semi-implicit semi-Lagrangian integration scheme of the spectral-transform IFS.

## 1 Introduction

Notwithstanding the achievements made over the last decades (Bauer et al., 2015), numerical weather prediction (NWP) faces the formidable challenge of resolving rather than parameterizing essential small-scale forcings and circulations in the multi-scale global flow – most notably processes associated with the surface, convective clouds, gravity waves, and troposphere–stratosphere interaction. While there is a need for advancement in many aspects of global NWP model infrastructures, prerequisites are the ability of the numerical model formulations to accurately predict atmospheric flows throughout the large-scale hydrostatic and small-scale nonhydrostatic regimes and to run efficiently on emerging and future high-performance computing (HPC) architectures.

**Published by Copernicus Publications on behalf of the European Geosciences Union.**

The spectral-transform (ST) – also known as pseudo-spectral – method was introduced in NWP following the work by Eliasen et al. (1970) and Orszag (1970). As a model representation entirely in spectral (i.e. wavenumber) space is impractical for NWP, the ST method maps between spectral and grid-point space in order to solve different parts of the governing equations in the space where the computations can be performed most efficiently. Typically, non-linear terms and the physics parameterizations are computed in grid-point space. Horizontal derivatives are computed in spectral space with formally high accuracy, as are linear terms of the discretized governing model equations – in particular, the constant-coefficient Helmholtz problem resulting from the semi-implicit time stepping (Robert et al., 1972) can be solved directly and accurately in spectral space. Facilitated by the ST method, the unconditional stability of the semi-implicit scheme combined with semi-Lagrangian (SL) advection in grid-point space permits very long time steps[1] and high efficiency (Ritchie et al., 1995; Temperton et al., 2001). In global models, the ST method typically uses a spherical harmonics representation in spectral space and (reduced, i.e. quasi-uniform) Gaussian grids (Hortal and Simmons, 1991; Wedi et al., 2015).

At ECMWF, the first forecast model using the ST method became operational in 1983, and the technique is still successfully applied today with the efficient SISL integration of the hydrostatic primitive equations in the Integrated Forecasting System (IFS) (Wedi et al., 2015). Recent advances helped to sustain the performance of the ST method (for details see Wedi et al., 2013, 2015; Wedi, 2014 and Sect. 2.2) and enabled real-time medium-range global weather forecasts at ECMWF with ≈ 9 km horizontal grid spacings in 2016 (Malardel et al., 2016). Furthermore, current research advanced the applicability of the ST method into the realm of global convection-permitting forecasts with kilometre-scale horizontal grid spacings (Wedi and Düben, 2017). While the viability of the ST method at ECMWF is ensured for the next decade, uncertainties concerning the scalability of the non-local high-volume parallel communications in the STs exist in the longer term (Wedi et al., 2013, 2015). These scalability issues could be exacerbated from the time when the increase in horizontal resolution makes the nonhydrostatic formulation based on the fully compressible equations (Bubnová et al., 1995; Bénard et al., 2010) essential. This is because the associated solution procedure in the IFS requires – at least in its current implementation – a predictor–corrector approach in the semi-implicit integration scheme that involves a considerably larger number of STs per model

time step (Bénard et al., 2010; Wedi et al., 2015). Furthermore, the large overlap regions between parallel distributed-memory partitions required in the SL scheme of IFS-ST can also be an issue in the longer term.

The uncertainties concerning the SISL integration based on the ST method with regard to emerging and future HPC architectures is one of the main reasons for ECMWF and its European partners to look into alternative nonhydrostatic, all-scale global model formulations and discretization schemes to be incorporated in the IFS. With this objective in mind, the Finite-Volume Module of the IFS (henceforth IFS-FVM) is under development at ECMWF (Smolarkiewicz et al., 2016, 2017; Kühnlein and Smolarkiewicz, 2017). An important property of the finite-volume (FV) method applied in IFS-FVM is a compact spatial discretization stencil in "grid-point" space, associated with a distributed-memory communication footprint that is predominantly local and performed using thin overlap regions with the nearest neighbours, in contrast to the non-local high-volume communications required in IFS-ST. In addition, advantages of the FV method are inherently local conservation and the ability to operate in complex, unstructured-mesh geometries. The lack of conservation is a common issue with standard SL schemes and a shortcoming in the current operational IFS. Conservation errors are presumed to contribute to significant (moisture and temperature) biases in the upper troposphere lower stratosphere region (Wedi et al., 2015), and a small but systematic drift in air mass and tracer fields may affect the forecast quality at longer (sub-)seasonal forecast ranges (Thuburn, 2008; Diamantakis and Flemming, 2014). The ability of FV methods to operate in complex, unstructured-mesh geometries is of high relevance to global NWP. In the global (spherical or spheroidal) domains, the FV technique provides ample freedom for implementing efficient quasi-uniform resolution meshes that circumvent the polar anisotropy of the classical regular longitude–latitude grids commonly employed with finite-difference (FD) discretization methods (Prusa et al., 2008; Wood et al., 2014). Flexibility with respect to the mesh is also important for implementing variable and/or adaptive resolution in atmospheric modelling systems, in which locally finer mesh spacings in sensitive regions (e.g. storm tracks) may provide an efficient way towards a more accurate representation of multi-scale interactions (Bacon et al., 2000; Weller et al., 2010; Kühnlein et al., 2012; Zarzycki et al., 2014).

By default, IFS-FVM employs 3-D semi-implicit integrators for the nonhydrostatic fully compressible equations (Smolarkiewicz et al., 2014). The all-scale integrators in IFS-FVM are conceptually akin to the semi-implicit schemes in IFS-ST but more general. In both formulations, accurate and robust integration with large time steps is achieved by 3-D implicit representation of fast acoustic and buoyant modes supported by the fully compressible equations. Furthermore, fully implicit representation of slow rotational modes is another common feature of both IFS-FVM and IFS-

---

[1]The SL schemes are subject to a topological realizability condition based on the Lipschitz number which is related to the flow deformation (Smolarkiewicz and Pudykiewicz, 1992; Cossette et al., 2014). However, in NWP this condition is typically much less restrictive than the advective CFL stability condition of Eulerian schemes; see e.g. Diamantakis and Magnusson (2016).

ST (Temperton, 2011; Smolarkiewicz et al., 2016). Although implicit time stepping is predominantly associated with computational stability, there are indications for favourable balance and accuracy in multi-scale flows (Knoll et al., 2003; Dörnbrack et al., 2005; Wedi and Smolarkiewicz, 2006). In contrast to IFS-ST, IFS-FVM's semi-implicit schemes do not rely on constant coefficients of the operator that is represented implicitly, as is required with the spectral space representation (Bénard et al., 2010). In IFS-ST, the operator that is represented implicitly results from linearization of the full non-linear governing equations about a horizontal reference state, and the semi-implicit integration then treats the non-linear residual (i.e. full non-linear equations minus the linear operator) explicitly (see Sect. 2.2). Constant coefficients effectively exclude orographic forcing from the linear operator of the semi-implicit scheme[2], leaving the associated effects solely to the explicit non-linear residual. Furthermore, in the constant-coefficient semi-implicit scheme different (i.e. split) boundary conditions are applied in the linear operator and the non-linear residual. Although still a research issue, the constant-coefficient semi-implicit scheme may incur reduced stability under more complex orography for future high-resolution forecasts in the nonhydrostatic regime.

At ECMWF, the reign of the ST method with the SISL integrators still continues, but future challenges, especially with respect to HPC, nonhydrostatic modelling, and complex orography, can be foreseen. The IFS-FVM represents an alternative dynamical core formulation that can complement IFS-ST with regard to these issues. However, to make IFS-FVM a useful option for global medium-range weather forecasting at ECMWF, it needs to be shown that the model formulation can provide (at least) comparable solution quality to the established IFS. In particular, a fundamental scientific question is whether a second-order FV method on the co-located meshes employed in IFS-FVM can sustain the accuracy of the ST method of IFS-ST. Another important question concerns the computational efficiency of IFS-FVM. At ECMWF and generally in NWP, tight constraints exist with regard to the runtime of the forecast models on the employed supercomputers. Therefore, we will evaluate the basic efficiency in terms of the time to solution of the current IFS-FVM formulation relative to the operational hydrostatic IFS-ST and its nonhydrostatic extension. In the present paper, these issues are investigated using relevant atmospheric flow benchmarks such as those defined in the context of the Dynamical Core Model Intercomparison Project (DCMIP; Ullrich et al., 2017). The DCMIP-2016 benchmarks involve large-scale hydrostatic and small-scale nonhydrostatic flows on the sphere and also emphasize the interaction of the dy-

namical core with selected parameterizations of sub-grid-scale physical processes. In the present paper IFS-FVM is verified against the proven IFS-ST at ECMWF for the baroclinic instability benchmark in the hydrostatic regime, considering specific configurations and parameterizations of interest at ECMWF. IFS-FVM also participates in the wider DCMIP-2016 model intercomparison, and this includes the nonhydrostatic supercell test case (Zarzycki et al., 2018).

The paper is organized as follows. Section 2 addresses the IFS-FVM and IFS-ST model formulations and juxtaposes their main formulation features. In particular, while Sect. 2.1 provides a description of the advanced semi-implicit finite-volume integration scheme of the novel IFS-FVM, Sect. 2.2 briefly summarizes the established IFS-ST. Furthermore, Sect. 2.3 discusses some basic aspects of the coupling to physics parameterizations, and Sect. 2.4 describes the common octahedral reduced Gaussian grid applied at ECMWF. Having described the model formulations, the IFS-FVM and IFS-ST benchmark simulations are then compared in Sect. 3. Section 4 concludes the paper.

## 2 IFS model formulations

The IFS comprises a comprehensive model infrastructure to perform data assimilation and to run deterministic and probabilistic global weather forecasts with various ranges and resolutions, supplemented with preprocessing and post-processing capabilities. The dynamical core lies at the heart of the NWP model infrastructure.

### 2.1 Finite-Volume Module of the IFS

IFS-FVM solves the deep-atmosphere[3], nonhydrostatic, fully compressible equations with a generalized height-based terrain-following vertical coordinate. Numerical integration of the governing equations employs a centred two-time-level semi-implicit scheme that provides unconditional stability in 3-D with respect to the fast acoustic and buoyant modes, as well as slower rotational modes (Smolarkiewicz et al., 2014, 2016). In Sect. 2.1.2, we extend the IFS-FVM semi-implicit integration to comprehensive moist-precipitating dynamics coupled to the IFS cloud physics parameterizations – this generalizes the simplified moist-precipitating dynamics with different cloud physics coupling described in Smolarkiewicz et al. (2017). The IFS-FVM semi-implicit integration is combined with non-oscillatory forward-in-time (NFT) Eulerian advection based on MPDATA (multidimensional positive definite advection transport algorithm) (Kühnlein and Smolarkiewicz, 2017). In the present work, new efficient horizontal–vertical split NFT advective transport schemes based on MPDATA are developed and applied (Sect. 2.1.2 and Appendix A). In addition, improved efficacy with the

---

[2]Including orography in the implicit part involves multiplications which are standardly performed in grid-point space in the context of the ST method. In principle, one could carry out the necessary multiplications in spectral space but this is usually avoided because of computational complexity.

[3]The shallow-atmosphere equations, the default in IFS-ST, are available by means of a simple switch $\gamma$ (Appendix C).

**Table 1.** Summary of the main formulation features of IFS-FVM and IFS-ST. For IFS-ST, information about the hydrostatic formulation and its nonhydrostatic extension is provided (see main text for description). Abbreviations are as follows: finite element (FE), finite difference (FD), spectral transform (ST), finite volume (FV), two time level (2-TL), semi-implicit (SI), iterative-centred implicit (ICI). A summary of variables is provided in Table D1.

| Model aspect | IFS-FVM | IFS-ST | IFS-ST (NH option) |
|---|---|---|---|
| Equation system | fully compressible | hydrostatic primitive | fully compressible |
| Prognostic variables | $\rho_{\mathrm{d}}, u, v, w, \theta', \varphi', r_{\mathrm{v}}, r_{\mathrm{l}}, r_{\mathrm{r}}, r_{\mathrm{i}}, r_{\mathrm{s}}$ | $\ln p_{\mathrm{s}}, u, v, T_{\mathrm{v}}, q_{\mathrm{v}}, q_{\mathrm{l}}, q_{\mathrm{r}}, q_{\mathrm{i}}, q_{\mathrm{s}}$ | $\ln \pi_{\mathrm{s}}, u, v, d_4, T_{\mathrm{v}}, \hat{q}, q_{\mathrm{v}}, q_{\mathrm{l}}, q_{\mathrm{r}}, q_{\mathrm{i}}, q_{\mathrm{s}}$ |
| Horizontal coordinates | $\lambda, \phi$ (lon–lat) | $\lambda, \phi$ (lon–lat) | $\lambda, \phi$ (lon–lat) |
| Vertical coordinate | generalized height | hybrid sigma–pressure | hybrid sigma–pressure |
| Horizontal discretization | unstructured finite volume (FV) | spectral transform (ST) | spectral transform (ST) |
| Vertical discretization | structured FD–FV | structured FE | structured FD–FE |
| Horizontal staggering | co-located | co-located | co-located |
| Vertical staggering | co-located | co-located | co-located, Lorenz |
| Horizontal grid | octahedral Gaussian or arbitrary | octahedral Gaussian | octahedral Gaussian |
| Time stepping scheme | 2-TL SI | 2-TL constant-coefficient SI | 2-TL constant-coefficient SI with ICI |
| Advection | conservative FV Eulerian | non-conservative SL | non-conservative SL |

Eulerian NFT MPDATA advection is sought by rigorous implementation of variable time stepping. The unstructured horizontal spatial discretization uses the median-dual FV approach of Szmelter and Smolarkiewicz (2010) combined with a structured-grid FD–FV approach in the vertical direction (Smolarkiewicz et al., 2016). In Sect. 2.1.3, we present a revised computationally efficient implementation of the median-dual FV approach. High efficacy also results from effective preconditioning of the Krylov-subspace solver for the Helmholtz problem arising in the semi-implicit time stepping of the fully compressible equations addressed in Sect. 2.1.2 and Appendix B. To facilitate interoperability between the different numerical methods in the IFS, IFS-FVM uses the median-dual FV mesh built about the nodes of the octahedral reduced Gaussian grid. However, the IFS-FVM numerical formulation is not restricted to this grid and offers capabilities towards broad classes of meshes including adaptivity (Szmelter and Smolarkiewicz, 2010; Kühnlein et al., 2012). The IFS-FVM employs flexible parallel data structures provided by ECMWF's Atlas library (Deconinck et al., 2017). The main model formulation features of IFS-FVM are summarized in Table 1 and shown alongside the corresponding IFS-ST properties discussed below in Sect. 2.2.

### 2.1.1 Governing equations

Building on the formulation of moist-precipitating dynamics described in Smolarkiewicz et al. (2017), the fully compressible equations considered in IFS-FVM are given as [TS1]

$$\frac{\partial \mathcal{G}\rho_{\mathrm{d}}}{\partial t} + \nabla \cdot (\boldsymbol{v}\mathcal{G}\rho_{\mathrm{d}}) = 0 \,, \tag{1a}$$

$$\frac{\partial \mathcal{G}\rho_{\mathrm{d}}\boldsymbol{u}}{\partial t} + \nabla \cdot (\boldsymbol{v}\mathcal{G}\rho_{\mathrm{d}}\boldsymbol{u}) =$$

$$\mathcal{G}\rho_{\mathrm{d}}\left[ -\theta_{\rho}\widetilde{\mathbf{G}}\nabla \varphi' + \boldsymbol{g}\left(1 - \frac{\vartheta}{\theta_{\rho\mathrm{a}}}\left(\theta_{\mathrm{a}} + \theta'\right)\right)\right.$$

$$\left. - \boldsymbol{f} \times \left(\boldsymbol{u} - \frac{\theta_{\rho}}{\theta_{\rho\mathrm{a}}}\boldsymbol{u}_{\mathrm{a}}\right) + \mathcal{M}' + \boldsymbol{P^u}\right], \tag{1b}$$

$$\frac{\partial \mathcal{G}\rho_{\mathrm{d}}\theta'}{\partial t} + \nabla \cdot (\boldsymbol{v}\mathcal{G}\rho_{\mathrm{d}}\theta') = \mathcal{G}\rho_{\mathrm{d}}\left[-\widetilde{\mathbf{G}}^{\mathrm{T}}\boldsymbol{u} \cdot \nabla \theta_{\mathrm{a}} + P^{\theta'}\right], \tag{1c}$$

$$\frac{\partial \mathcal{G}\rho_{\mathrm{d}}\, r_k}{\partial t} + \nabla \cdot (\boldsymbol{v}\mathcal{G}\rho_{\mathrm{d}}\, r_k) = \mathcal{G}\rho_{\mathrm{d}}P^{r_k}\,, \quad r_k = r_{\mathrm{v}}, r_{\mathrm{l}}, r_{\mathrm{r}}, r_{\mathrm{i}}, r_{\mathrm{s}} \tag{1d}$$

$$\varphi' = c_{\mathrm{pd}}\left[\left(\frac{R_{\mathrm{d}}}{p_0}\rho_{\mathrm{d}}\,\theta\,(1 + r_{\mathrm{v}}/\varepsilon)\right)^{R_{\mathrm{d}}/c_{\mathrm{vd}}} - \pi_{\mathrm{a}}\right], \tag{1e}$$

which describe the conservation laws of dry mass (Eq. 1a), momentum (Eq. 1b), dry entropy (Eq. 1c), and water substance (Eq. 1d). A summary of variables and physical constants is provided in Tables D1 and D2, respectively. Dependent variables in Eqs. (1a)–(1e) are dry density $\rho_{\mathrm{d}}$, three-dimensional physical velocity vector $\boldsymbol{u} = [u, v, w]^{\mathrm{T}}$, potential temperature perturbation $\theta'$, and a modified Exner pressure perturbation $\varphi' \equiv c_{\mathrm{pd}}\pi'$, as well as the water substance mixing ratios $r_k = \rho_k/\rho_{\mathrm{d}}$ (i.e. the ratio of the density of the individual water substance category $\rho_k$ to the density of dry air $\rho_{\mathrm{d}}$); with the current cloud parameterization of the IFS (Forbes et al., 2010), five categories for water substance are considered (vapour $r_{\mathrm{v}}$, liquid $r_{\mathrm{l}}$, rain $r_{\mathrm{r}}$, ice $r_{\mathrm{i}}$, snow $r_{\mathrm{s}}$), each described by the respective PDE (Eq. 1d). An additional prognostic equation for cloud fraction $\Lambda_{\mathrm{a}}$ employed with the IFS cloud parameterization is implemented as

$$\frac{\partial \mathcal{G}\rho_{\mathrm{d}}\,\Lambda_{\mathrm{a}}}{\partial t} + \nabla \cdot (\boldsymbol{v}\mathcal{G}\rho_{\mathrm{d}}\,\Lambda_{\mathrm{a}}) = \mathcal{G}\rho_{\mathrm{d}}P^{\Lambda_{\mathrm{a}}}\,. \tag{2}$$

Furthermore, the thermodynamic variables are related by the gas law (Eq. 1e); the Exner pressure is $\pi = (p/p_0)^{R_{\mathrm{d}}/c_{\mathrm{pd}}}$ and the potential temperature is $\theta = T/\pi$, where $p$ is the total pressure, $p_0 \equiv 10^5$ Pa, and $T$ is the (absolute) temperature. In Sect. 2.1.2, the fully compressible system (1a)–(1e) [TS2] is augmented with the prognostic Helmholtz Eq. (17) derived from the advective form of the gas law (Eq. 1e) for the im-

plicit (rather than explicit) solution with respect to the Exner pressure perturbation $\varphi'$.

A quantity which appears in various right-hand-side (RHS) terms of the momentum Eq. (1b) is the density potential temperature $\theta_\rho = \theta\,\vartheta$, where $\vartheta \equiv (1 + r_v/\varepsilon)/(1 + r_t)$ (Emanuel, 1994) with $\varepsilon = R_d/R_v$, and the total water mixing ratio $r_t$ represents the sum over all the individual mixing ratios:

$$r_t = \sum_k r_k = r_v + r_l + r_r + r_i + r_s \,. \tag{3}$$

The multiplying factor $\vartheta$ appears explicitly in the buoyancy term of Eq. (1b) in order to expose the potential temperature perturbation $\theta'$ for implicit coupling to the thermodynamic Eq. (1c) (see Sect. 2.1.2). Note that a high-order approximation of the buoyancy term applied in Kurowski et al. (2014) and Smolarkiewicz et al. (2017) for consistency with soundproof models at mesoscales has been replaced here by the full, unabbreviated form essential for the accurate representation of moist-precipitating dynamics at planetary scales (Sect. 3).

Another important aspect of the governing Eqs. (1a)–(1e) is the underlying perturbation form. The perturbations of potential temperature $\theta' = \theta - \theta_a$ and Exner pressure $\pi' = \pi - \pi_a$ correspond to deviations from an ambient state (denoted by the subscript "a") that satisfies a general balanced subset of Eqs. (1a)–(1e). A straightforward example applied in Sect. 3 of this paper is a stationary atmosphere $\boldsymbol{u}_a(\boldsymbol{x})$, $\theta_a(\boldsymbol{x})$, $\pi_a(\boldsymbol{x})$, $r_{va}(\boldsymbol{x})$ in thermal wind balance, which in terms of the momentum Eq. (1b) is

$$0 = -\theta_{\rho a}\widetilde{\mathbf{G}}\nabla\varphi_a - \boldsymbol{g} - \boldsymbol{f} \times \boldsymbol{u}_a + \boldsymbol{\mathcal{M}}(\boldsymbol{u}_a) \,, \tag{4}$$

where $\varphi_a \equiv c_{pd}\pi_a$ and $\theta_{\rho a} = \theta_a (1 + r_{va}/\varepsilon)/(1 + r_{va})$. The perturbation form (1a)–(1e)[TS3] is analytically equivalent to the fully compressible equations for full variables but has favourable properties for numerical integration; see Sect. 2.1.2 and Prusa et al. (2008) and Smolarkiewicz et al. (2014). Moreover, there are other analytically equivalent perturbation forms of the fully compressible equations implemented in IFS-FVM, which may differ in the degree of implicitness permitted in the integration (Smolarkiewicz et al., 2019). The optimal specification of ambient states for NWP and climate modelling is ongoing research. In general, ambient states can be as simple as stationary vertical profiles in hydrostatic balance, but bespoke definitions designed for a particular class of applications can substantially benefit the accuracy and robustness of the model integration.

The governing equations in Eqs. (1a)–(2)[TS4] are formulated with respect to generalized curvilinear coordinates embedded in a geospherical framework. At the most elementary level, the generalized curvilinear coordinate formulation can be used to implement fixed terrain-following levels with appropriate boundary conditions, but the model formulation optionally permits quite general moving meshes in the vertical

and the horizontal directions. Symbols associated with the geometric aspects of the model are the transformed curvilinear coordinates $\boldsymbol{x} = [x, y, z]^T$, the 3-D nabla operator $\nabla$ with respect to $\boldsymbol{x}$, the Jacobian $\mathcal{G}$ of the coordinate transformations (i.e. the square root of the determinant of the metric tensor), a matrix of metric coefficients $\widetilde{\mathbf{G}}$, its transpose $\widetilde{\mathbf{G}}^T$, and the contravariant velocity $\boldsymbol{v} = \dot{\boldsymbol{x}} = \widetilde{\mathbf{G}}^T\boldsymbol{u} + \boldsymbol{v}^g$ where $\boldsymbol{v}^g \equiv \partial\boldsymbol{x}/\partial t$ is the mesh velocity; see Prusa and Smolarkiewicz (2003) and Kühnlein et al. (2012) for extended discussion. Following Smolarkiewicz et al. (2017), further symbols on the RHS of the momentum Eq. (1b) denote the gravity vector $\boldsymbol{g} = [0, 0, -g]^T$, the Coriolis parameter $\boldsymbol{f} \equiv 2\boldsymbol{\Omega}$ with $\boldsymbol{\Omega}$ the angular velocity vector of the Earth's rotation, and $\boldsymbol{\mathcal{M}}'$ subsumes the metric forces due to the curvature of the sphere:

$$\boldsymbol{\mathcal{M}}'\big(\boldsymbol{u}, \boldsymbol{u}_a, \theta_\rho/\theta_{\rho a}\big) = \boldsymbol{\mathcal{M}}(\boldsymbol{u}) - (\theta_\rho/\theta_{\rho a})\boldsymbol{\mathcal{M}}(\boldsymbol{u}_a) \,. \tag{5}$$

Details about the specification of the curvilinear space, as well as explicit expressions of the Coriolis term $-\boldsymbol{f} \times \boldsymbol{u}$, the metric forces $\boldsymbol{\mathcal{M}}$, and the gravity $g$ under shallow- or deep-atmosphere equations, are given in Appendix C. Last but not least, the symbols $\boldsymbol{P^u} = [P^u, P^v, P^w]^T$, $P^{\theta'}$, $P^{r_k}$ on the RHS of Eqs. (1a)–(1e) denote the respective forcings from physics parameterizations.

Note that additional RHS terms not explicitly provided in the governing Eqs. (1a)–(1e) may describe Rayleigh-type damping and/or Laplacian diffusion applied especially to model wave-absorbing layers at the domain boundaries; see e.g. Prusa and Smolarkiewicz (2003) and Klemp et al. (2008).

### 2.1.2 Semi-implicit numerical integration

To facilitate a compact description of the integration scheme, each of the governing equations of the system (1a)–(1e)[TS5] is accommodated in a generalized conservation law form:

$$\frac{\partial G\Psi}{\partial t} + \nabla \cdot (\boldsymbol{V}\Psi) = G\left(\mathcal{R}^\Psi + P^\Psi\right) \,, \tag{6}$$

in which $\Psi$ denotes the prognostic model variable, $\boldsymbol{V}$ the advector, $G$ a generalized density, $P^\Psi$ represents the forcing from physics parameterization, and $\mathcal{R}^\Psi$ the remaining right-hand side[4]; see Table 2 for the respective specifications of $\Psi$, $\boldsymbol{V}$, $G$. The homogeneous mass continuity Eq. (1a) is a particular case of Eq. (6) and plays a fundamental role for the conservative advective transport of all other scalar variables. Note that because of the mass continuity Eq. (1a), the other scalar conservation laws (Eq. 6) are equivalent to the Lagrangian form $d\Psi/dt = \mathcal{R}^\Psi + P^\Psi$, where $d/dt = \partial/\partial t + \boldsymbol{v} \cdot \nabla$ represents the total derivative.

Building on the earlier works by Smolarkiewicz et al. (2014, 2016), the two-time-level numerical integrators of IFS-FVM for Eq. (6) can be

---

[4]As an example, $\mathcal{R}^{\theta'} \equiv -\widetilde{\mathbf{G}}^T\boldsymbol{u} \cdot \nabla\theta_a$ when considering Eq. (1c).

subsumed in the following template scheme:

$$\Psi_{\boldsymbol{i}}^{n+1} = \mathcal{A}_{\boldsymbol{i}}(\widetilde{\Psi}, \boldsymbol{V}^{n+1/2}, G^n, G^{n+1}, \delta t) + b^\Psi \, \delta t \, \mathcal{R}^\Psi|_{\boldsymbol{i}}^{n+1}$$
$$\equiv \widehat{\Psi}_{\boldsymbol{i}} + b^\Psi \, \delta t \, \mathcal{R}^\Psi|_{\boldsymbol{i}}^{n+1} \, , \tag{7}$$

where

$$\widetilde{\Psi} = \Psi^n + a^\Psi \, \delta t \, \mathcal{R}^\Psi|^n + \delta t \, P^\Psi|^n \, , \tag{8}$$

and $\widehat{\Psi}_{\boldsymbol{i}} = \mathcal{A}_{\boldsymbol{i}}(\widetilde{\Psi}, \boldsymbol{V}^{n+1/2}, G^n, G^{n+1}, \delta t)$ in Eq. (7) symbolizes a flux-form Eulerian NFT advective transport scheme based on MPDATA, as described in Kühnlein and Smolarkiewicz (2017) and Appendix A of the present paper. The $n, n+1/2, n+1$ indices denote full and half-time levels, and $\delta t = t^{n+1} - t^n$ is the time step increment of the semi-implicit dynamics. The vector index $\boldsymbol{i} = (k, i)$ marks the node positions $k$ and $i$ of the vertical and horizontal computational mesh, respectively, thereby revealing the 3-D co-located spatial arrangement of all dependent variables underlying IFS-FVM's discretization. The definitions of the weights $a^\Psi$ and $b^\Psi$ for the incorporation of the right-hand-side terms $\mathcal{R}^\Psi$ at $t^n$ and $t^{n+1}$, respectively, are given in Table 2. Apart from the incorporation of the physics parameterization $P^\Psi$, the semi-implicit scheme (7) with weights $a^\Psi \equiv b^\Psi \equiv 0.5$ is fully congruent with the second-order trapezoidal-rule trajectory integral of the corresponding ordinary differential equation

$$\frac{\mathrm{d}\Psi}{\mathrm{d}t} = \mathcal{R}^\Psi \tag{9}$$

of Eq. (6) (Smolarkiewicz and Margolin, 1993). Due to the congruency of Eq. (7) with the ODE (Eq. 9), the advection operator $\widehat{\Psi}_{\boldsymbol{i}}$ may equally represent a second-order accurate semi-Lagrangian scheme (Smolarkiewicz et al., 2014). In terms of the coupling to physics parameterizations formally incorporated with first-order accuracy, the associated forcing $P^\Psi|^n = P^\Psi(t_{\mathrm{phys}}, \Delta t_{\mathrm{phys}})$ can optionally be evaluated with an equal or longer time step $\Delta t_{\mathrm{phys}} = N_s \delta t$ (with integer $N_s = 1, 2, 3, \ldots$) than $\delta t$ applied in Eq. (7); see Sect. 2.3 about physics–dynamics coupling.

There are two alternative implementations of the NFT advective transport scheme $\mathcal{A}_{\boldsymbol{i}}(\widetilde{\Psi}, \boldsymbol{V}^{n+1/2}, G^n, G^{n+1}, \delta t)$ based on MPDATA. First, the standard MPDATA formulations for integrating the fully compressible equations in the horizontally unstructured vertically structured discretization framework of IFS-FVM are provided in Kühnlein and Smolarkiewicz (2017). These schemes are fully multidimensional (i.e. unsplit), equipped with non-oscillatory enhancement (Smolarkiewicz and Grabowski, 1990; Smolarkiewicz and Szmelter, 2005), and qualify for implicit large-eddy simulation (ILES) of high-Reynolds-number atmospheric flows; e.g. Domaradzki et al. (2003), Piotrowski et al. (2009), and Smolarkiewicz et al. (2013). Secondly, a more efficient horizontal–vertical split advective transport scheme based on MPDATA has been developed and is outlined in Appendix A. All IFS-FVM results presented in Sect. 3 were obtained

with this horizontal–vertical split scheme. Note that the basic unsplit and horizontal–vertical split MPDATA schemes are second-order accurate given the advector $\boldsymbol{V}^{n+1/2}$ at the intermediate time level $t^{n+1/2}$ provided as an $\mathcal{O}(\delta t^2)$ estimate, which is explained below.

Given the preceding discussion, the semi-implicit solution procedure of the governing system (1a)–(1e) [TS6] proceeds from an atmospheric state at $t^n$ to a state at $t^{n+1}$ as described in the following. The solution procedure commences with the integration of the mass continuity Eq. (1a) as

$$\rho_{\mathrm{d}\,\boldsymbol{i}}^{n+1} = \mathcal{A}_{\boldsymbol{i}}(\rho_{\mathrm{d}}^n, (\boldsymbol{v}\mathcal{G})^{n+1/2}, \mathcal{G}^n, \mathcal{G}^{n+1}, \delta t) \, , \tag{10}$$

which straightforwardly returns the updated density $\rho_{\mathrm{d}\,\boldsymbol{i}}^{n+1}$. The $\mathcal{O}(\delta t^2)$ estimate for the advector $(\boldsymbol{v}\mathcal{G})^{n+1/2}$ in Eq. (10) is implemented here by linear extrapolation of the advective velocities from the previous time levels $t^{n-1}$ and $t^n$; see Appendix A of Kühnlein et al. (2012) for the procedure accounting for a variable time step $\delta t$. In addition, the algorithm (10) defines the advector $\boldsymbol{V}^{n+1/2} \equiv (\boldsymbol{v}\mathcal{G}\rho_{\mathrm{d}})^{n+1/2}$ as face-normal mass fluxes to the dual cell $(\boldsymbol{v}\mathcal{G}\rho_{\mathrm{d}})^\perp|^{n+1/2}$ for the advective transport of all other scalar variables (see Table 2), a common approach to enable mass-compatible and monotonic solutions; see Smolarkiewicz et al. (2016) and Kühnlein and Smolarkiewicz (2017) for a discussion in the context of IFS-FVM.

Given the tendencies from physics parameterization $P^{\theta'}$, $P^u$, $P^v$, $P^{r_k}$, $P^{\Lambda_{\mathrm{a}}}$ formally evaluated at $t^n$ and the advective transport of all scalar variables $\widehat{\theta'}_{\boldsymbol{i}}$, $\widehat{\boldsymbol{u}}_{\boldsymbol{i}} \equiv (\widehat{u}_{\boldsymbol{i}}, \widehat{v}_{\boldsymbol{i}}, \widehat{w}_{\boldsymbol{i}})$, $r_{k\,\boldsymbol{i}}^{n+1} \equiv \widehat{r}_{k\,\boldsymbol{i}}$, $\Lambda_{\mathrm{a}\,\boldsymbol{i}}^{n+1} \equiv \widehat{\Lambda}_{\mathrm{a}\,\boldsymbol{i}}$ – with the advected water content mixing ratios $\widehat{r}_{k\,\boldsymbol{i}}$ and cloud fraction $\widehat{\Lambda}_{\mathrm{a}\,\boldsymbol{i}}$ already representing the final solutions at $t^{n+1}$ – the scheme (7)–(8) for the thermodynamic Eq. (1c) and momentum Eq. (1b) is implemented as

$$\theta'_{\boldsymbol{i}} = \widehat{\theta'}_{\boldsymbol{i}} - 0.5\delta t \left[ \widetilde{\mathbf{G}}^{\mathrm{T}} \boldsymbol{u} \cdot \nabla \theta_{\mathrm{a}} \right]_{\boldsymbol{i}} \tag{11a}$$

$$\boldsymbol{u}_{\boldsymbol{i}} = \widehat{\boldsymbol{u}}_{\boldsymbol{i}} + 0.5\delta t \left[ -\theta_\rho^\star \widetilde{\mathbf{G}} \nabla \varphi' + \boldsymbol{g} \left( 1 - \frac{\vartheta^{n+1}}{\theta_{\rho\mathrm{a}}} \left( \theta_{\mathrm{a}} + \theta' \right) \right) \right.$$
$$\left. - \boldsymbol{f} \times \left( \boldsymbol{u} - \frac{\theta_\rho^\star}{\theta_{\rho\mathrm{a}}} \boldsymbol{u}_{\mathrm{a}} \right) + \boldsymbol{\mathcal{M}}'\left( \boldsymbol{u}^\star, \boldsymbol{u}_{\mathrm{a}}, \frac{\theta_\rho^\star}{\theta_{\rho\mathrm{a}}} \right) \right]_{\boldsymbol{i}} \, , \tag{11b}$$

where

$$\theta_\rho^\star = \theta^\star \vartheta^{n+1} \equiv \frac{\theta^\star (1 + r_{\mathrm{v}}^{n+1}/\varepsilon)}{(1 + r_t^{n+1})} \, . \tag{12}$$

Due to the presence of non-linear terms, the discrete system (11a)–(11b) [TS7] is executed iteratively (Smolarkiewicz and Dörnbrack, 2008; Smolarkiewicz et al., 2014), with lagged quantities from the previous iteration denoted by the superscript $\star$. Typically, it uses one corrective iteration, which results in a predictor–corrector approach. The predictor of Eqs. (11a)–(11b) alone is second-order accurate, and the predictor–corrector already closely approximates the corresponding trapezoidal integral (Smolarkiewicz and Szmelter, 2009;

**Geosci. Model Dev., 12, 1–25, 2019** **www.geosci-model-dev.net/12/1/2019/**

**Table 2.** Specification of prognostic model variables and corresponding parameters in the template scheme (7)–(8). Columns represent the dependent variable $\Psi$, the advector $V$, a generalized density $G$, and $a^\Psi$, $b^\Psi$ the weights for the incorporation of the RHS forcings $\mathcal{R}^\Psi$. The rightmost column refers to the governing equation for each dependent variable $\Psi$.

| Variable | $\Psi$ | $V$ | $G$ | $a^\Psi$ | $b^\Psi$ | Eq. |
|---|---|---|---|---|---|---|
| Dry density | $\rho_\mathrm{d}$ | $v\mathcal{G}$ | $\mathcal{G}$ | – | – | (1a) |
| Zonal physical velocity | $u$ | $v\mathcal{G}\rho_\mathrm{d}$ | $\mathcal{G}\rho_\mathrm{d}$ | 0.5 | 0.5 | (1b) |
| Meridional physical velocity | $v$ | $v\mathcal{G}\rho_\mathrm{d}$ | $\mathcal{G}\rho_\mathrm{d}$ | 0.5 | 0.5 | (1b) |
| Vertical physical velocity | $w$ | $v\mathcal{G}\rho_\mathrm{d}$ | $\mathcal{G}\rho_\mathrm{d}$ | 0.5 | 0.5 | (1b) |
| Potential temperature perturbation | $\theta'$ | $v\mathcal{G}\rho_\mathrm{d}$ | $\mathcal{G}\rho_\mathrm{d}$ | 0.5 | 0.5 | (1c) |
| Water vapour mixing ratio | $r_\mathrm{v}$ | $v\mathcal{G}\rho_\mathrm{d}$ | $\mathcal{G}\rho_\mathrm{d}$ | – | – | (1d) |
| Liquid water mixing ratio | $r_\mathrm{l}$ | $v\mathcal{G}\rho_\mathrm{d}$ | $\mathcal{G}\rho_\mathrm{d}$ | – | – | (1d) |
| Rain water mixing ratio | $r_\mathrm{r}$ | $v\mathcal{G}\rho_\mathrm{d}$ | $\mathcal{G}\rho_\mathrm{d}$ | – | – | (1d) |
| Ice mixing ratio | $r_\mathrm{i}$ | $v\mathcal{G}\rho_\mathrm{d}$ | $\mathcal{G}\rho_\mathrm{d}$ | – | – | (1d) |
| Snow mixing ratio | $r_\mathrm{s}$ | $v\mathcal{G}\rho_\mathrm{d}$ | $\mathcal{G}\rho_\mathrm{d}$ | – | – | (1d) |
| Cloud fraction | $\Lambda_\mathrm{a}$ | $v\mathcal{G}\rho_\mathrm{d}$ | $\mathcal{G}\rho_\mathrm{d}$ | – | – | (2) |
| Exner pressure perturbation | $\varphi'$ | $v\mathcal{G}\rho_\mathrm{d}$ | $\mathcal{G}\rho_\mathrm{d}$ | $(1-\alpha)$ | $\alpha$ | (17) |

Smolarkiewicz et al., 2014, 2019). With the respective prognostic model variables $\theta'$ and $u$ in Eqs. (11a)–(11b) the $n+1$ time level index has been dropped, but the $n+1$ time level index is retained with the coefficient $\vartheta^{n+1}$ defined in Eq. (12) that is composed of the already completed $r_k^{n+1}$. At the beginning of the iterative execution of Eqs. (11a)–(11b), a first guess $\theta^0$ for $\theta^\star$ is provided as the explicit solution of full potential temperature:

$$\theta_i^0 = \widehat{\theta_i} = \mathcal{A}_i\left(\theta^n + \delta t\, P^{\theta'}|^n, (v\mathcal{G}\rho_\mathrm{d})^\perp|^{n+1/2}, (\mathcal{G}\rho_\mathrm{d})^n, \right.$$
$$\left. (\mathcal{G}\rho_\mathrm{d})^{n+1}, \delta t\right),\tag{13}$$

whereas a first guess $u^0$ for $u^\star$ is prescribed by linear extrapolation of $u$ from the $t^{n-1}$ and $t^n$ time levels (again taking into account the variable time step $\delta t$).

From Eqs. (11a)–(11b), closed-form expressions for the discrete velocity update are derived by eliminating Eq. (11a) for $\theta_i'$ in the buoyancy term of Eq. (11b) – thereby implementing 3-D fully implicit treatment of buoyant modes in a moist-precipitating atmosphere – and gathering the terms with linear dependence on $u_i$ on the left-hand side (LHS), which results in

$$u + 0.5\delta t\, f \times u - (0.5\delta t)^2 \frac{g}{\theta_\mathrm{a}} \widetilde{G}^\mathrm{T} u \cdot \nabla\theta_\mathrm{a} =$$
$$\widehat{u} - 0.5\delta t\left[\frac{g}{\theta_{\rho\mathrm{a}}}\left(\theta_{\rho\mathrm{a}} - \vartheta^{n+1}\theta_\mathrm{a} - \vartheta^{n+1}\widehat{\theta'}\right) - f \times \frac{\theta_\rho^\star}{\theta_{\rho\mathrm{a}}} u_\mathrm{a}\right.$$
$$\left. - \mathcal{M}'\left(u^\star, u_\mathrm{a}, \frac{\theta_\rho^\star}{\theta_{\rho\mathrm{a}}}\right)\right] - 0.5\delta t\, \theta_\rho^\star \widetilde{G}\nabla\varphi'$$
$$\equiv \widehat{\widehat{u}} - 0.5\delta t\, \theta_\rho^\star \widetilde{G}\nabla\varphi'.\tag{14}$$

As all terms are co-located, the spatial mesh vector index $i$ has been omitted in Eq. (14). The RHS of Eq. (14) is composed of all explicitly known terms, summarized as $\widehat{\widehat{u}}$, and the pressure gradient term with the lagged coefficient $\theta_\rho^\star$. Defining $\mathbf{L}$ as the linear operator acting on $u$ on the LHS

and $\mathbf{L}^{-1}$ its inverse, Eq. (14) can be symbolized as

$$\mathbf{L}u = \widehat{\widehat{u}} - 0.5\delta t\, \theta_\rho^\star \widetilde{G}\nabla\varphi',\tag{15}$$

and

$$u = \check{u} - \mathbf{C}\nabla\varphi',\tag{16}$$

respectively, where $\check{u} = \mathbf{L}^{-1}\widehat{\widehat{u}}$, and $\mathbf{C} = \mathbf{L}^{-1}0.5\,\delta t\, \theta_\rho^\star \widetilde{G}$ denotes a $3 \times 3$ matrix of known coefficients. The solution algorithm presented up to Eq. (16) still requires the Exner pressure perturbation $\varphi'$ to be specified. A straightforward computation may employ the gas law (Eq. 1e) using the updated variables $\rho_\mathrm{d}$, $\theta$, and $r_\mathrm{v}$. However, the resulting 3-D explicit acoustic integration is subject to very small time steps (in order to maintain numerical stability) and thus inefficient for NWP. Therefore, a final step in IFS-FVM's numerical solution procedure is to augment the fully compressible Eqs. (1a)–(1e) with an auxiliary 3-D implicit boundary value problem for the pressure perturbation variable $\varphi'$ (Smolarkiewicz et al., 2014). The formulation of this implicit boundary value problem originates from the advective (or Lagrangian) form of the gas law (Eq. 1e) combined with the respective advective forms of the mass continuity Eq. (1a), thermodynamic Eq. (1c), and water vapour Eq. (1d) equations; see Smolarkiewicz et al. (2014, 2017) for further details. The governing equation for $\varphi'$ is then implemented in conservation form consistent with Eqs. (1a)–(1e) and reads

$$\frac{\partial \mathcal{G}\rho_\mathrm{d}\varphi'}{\partial t} + \nabla\cdot\left(v\mathcal{G}\rho_\mathrm{d}\varphi'\right) =$$
$$\mathcal{G}\rho_\mathrm{d}\left[-\frac{R_\mathrm{d}}{c_{\mathrm{vd}}}\frac{\varphi}{\mathcal{G}}\nabla\cdot(\mathcal{G}\,\widetilde{G}^\mathrm{T}u) - \frac{1}{\mathcal{G}\rho_\mathrm{d}}\nabla\cdot(\mathcal{G}\rho_\mathrm{d}\widetilde{G}^\mathrm{T}u\,\varphi_\mathrm{a})\right.$$
$$\left. + \frac{\varphi_\mathrm{a}}{\mathcal{G}\rho_\mathrm{d}}\nabla\cdot(\mathcal{G}\rho_\mathrm{d}\widetilde{G}^\mathrm{T}u) + \frac{R_\mathrm{d}}{c_{\mathrm{vd}}}\varphi\,\Pi\right],\tag{17}$$

where $\varphi = \varphi_{\mathrm{a}} + \varphi'$ and

$$\Pi = \left( \frac{P^{\theta'}}{\theta} + \frac{P^{r_{\mathrm{v}}}/\varepsilon}{1 + r_{\mathrm{v}}/\varepsilon} \right) . \tag{18}$$

We interpret Eq. (17) in terms of the generalized transport Eq. (6) for $\Psi \equiv \varphi'$ (see Table 2), while setting

$$P^{\varphi'} = (R_{\mathrm{d}}/c_{\mathrm{vd}})\, \varphi\, \Pi \;, \tag{19}$$

and $\mathcal{R}^{\varphi'}$ as the remaining RHS consisting of the three divergence operators. Importantly, the $P^{\varphi'}$ given by Eq. (19) describes the pressure adjustment from the physics parameterization tendencies $P^{\theta'}$ and $P^{r_{\mathrm{v}}}$. We have implemented the integration of Eq. (17) using the template semi-implicit scheme (7) with parameters $a^{\varphi'} = (1-\alpha)$ and $b^{\varphi'} = \alpha$, where $\alpha \in [0.5, 1.0]$. In the limit $\alpha = 1.0$, the template Eq. (7) represents the first-order backward Euler scheme with regard to $\mathcal{R}^{\varphi'}$. For $\alpha = 0.5$, the template Eq. (7) becomes the second-order trapezoidal scheme; in practice we may use weak off-centring (e.g. $\alpha = 0.51$) for regularization. For any specification of $\alpha$, coupling with the solution procedure of the fully compressible Eqs. (1a)–(1e) is implemented through Eq. (16) that enters into the three occurrences of $\boldsymbol{u}$ on the RHS of Eq. (17) in $\mathcal{R}^{\varphi'}|^{n+1}$. Furthermore, as indicated in Eq. (7), the (explicit) forward Euler scheme is used with regard to the forcing $P^{\varphi'}$. Reorganizing terms finally yields the elliptic Helmholtz equation for the pressure perturbation variable $\varphi'$ at the future time level $t^{n+1}$, which can be written compactly as

$$0 = -\sum_{\ell=1}^{3} \left( \frac{A_\ell^\star}{\zeta_\ell} \nabla \cdot \zeta_\ell\, \widetilde{\mathbf{G}}^{\mathrm{T}}(\check{\boldsymbol{u}} - \mathbf{C}\nabla\varphi') \right) - B^\star(\varphi' - \widehat{\varphi'})$$
$$\equiv \mathcal{L}(\varphi') - R \;, \tag{20}$$

where the spatial grid index $\boldsymbol{i}$ has been omitted again. The summation $\ell$ in Eq. (20) is over the three divergence operators on the RHS of Eq. (17). The symbolic notations $\mathcal{L}(\varphi')$ and $R$ refer to the implicit and explicit parts of the equation, respectively. The coefficients $A_\ell^\star$, $\zeta_\ell$, and $B^\star$ are defined accordingly as

$$A_1^\star = 1, \quad A_2^\star = A_3^\star = \frac{c_{\mathrm{vd}}}{R_{\mathrm{d}}} \frac{\varphi_{\mathrm{a}}}{\varphi}\;, \quad \zeta_1 = \mathcal{G}, \quad \zeta_2 = \mathcal{G}\rho_{\mathrm{d}}\varphi_{\mathrm{a}},$$

$$\zeta_3 = \mathcal{G}\rho_{\mathrm{d}}, \quad B^\star = \frac{1}{\alpha\, \delta t} \frac{c_{\mathrm{vd}}}{R_{\mathrm{d}}\,\varphi}\;, \tag{21}$$

and $\widehat{\varphi'} = \mathcal{A}(\varphi'^n + (1-\alpha)\, \delta t\, \mathcal{R}^{\varphi'}|^n + \delta t\, P^{\varphi'}|^n, (\boldsymbol{v}\mathcal{G}\rho_{\mathrm{d}})^\perp|^{n+1/2}, (\mathcal{G}\rho_{\mathrm{d}})^n, (\mathcal{G}\rho_{\mathrm{d}})^{n+1}, \delta t)$. The Helmholtz Eq. (20) extends the implementations in Smolarkiewicz et al. (2016, 2017) and Kühnlein and Smolarkiewicz (2017) that all used the backward Euler scheme $\alpha = 1$. The 3-D elliptic boundary value problem (Eq. 20) is solved using a nonsymmetric preconditioned generalized conjugate residual (GCR) approach; see Smolarkiewicz and Szmelter (2011) for a recent discussion and Smolarkiewicz and Margolin (2000) and

Smolarkiewicz et al. (2004) for tutorials. The crux of the preconditioning is the direct inversion of the vertical part of the problem that dramatically reduces the condition number of the full linear operator $\mathcal{L}$ and enables rapid convergence of the GCR solver. The preconditioner $\mathcal{P} \approx \mathcal{L}$ discards the off-diagonal entries of the matrix $\widetilde{\mathbf{G}}^{\mathrm{T}}\mathbf{C}$. Inversion of $\mathcal{P}$ then utilizes a weighted line Jacobi method explained in Appendix B. In the GCR solver and the preconditioner $\mathcal{P}$, the coefficients $A_2^\star$, $A_3^\star$, and $B^\star$ in Eq. (21) depend on $\varphi = \varphi_{\mathrm{a}} + \varphi'$, lagged behind in the outer iteration of Eq. (11b). Importantly, in the predictor step of the outer iteration, an improved first guess for $\varphi'$, can be achieved by employing the explicit forward Euler scheme for Eq. (17), obtained by setting $\alpha = 0$ in Eq. (7).

As far as the adiabatic dynamics is concerned, the $\mathcal{O}(\delta t^2)$ integration of Eq. (17) with the first-order backward Euler scheme $\alpha = 1$ maintains second-order accuracy of all variables except $\varphi'$ over a single time step because $\varphi'$ enters the pressure gradient term of the momentum equation with the factor $0.5\delta t$, hence resulting in an $O(\delta t^3)$ integral. The accumulation of first-order errors in $\varphi'$ can be mitigated by solving Eq. (17) with the weakly off-centred trapezoidal scheme using $\alpha = 0.51$; see Benacchio et al. (2014) for alternative design and pertinent discussion. All IFS-FVM results presented in Sect. 3 were obtained with the backward Euler scheme $\alpha = 1$, as this has been the default in the earlier implementations. However, using the extended-range forecast of the baroclinic instability benchmark, we will demonstrate that either $\alpha = 1$ or $\alpha = 0.51$ shows the same close agreement with IFS-ST. Moreover, both choices $\alpha = 1$ or $\alpha = 0.51$ provide essentially identical computational performance.

Overall, the 3-D implicit scheme with respect to $\varphi'$ permits time steps equivalent to soundproof models (Kurowski et al., 2014; Smolarkiewicz et al., 2014; Kühnlein and Smolarkiewicz, 2017). Together with the full 3-D implicit incorporation of buoyant and rotational modes described above, the semi-implicit integration is unconditionally stable with respect to all waves supported by the fully compressible Eqs. (1a)–(1e), and thus the semi-implicit model time step $\delta t$ can be selected according to the stability of the advective transport scheme $\mathcal{A}_{\boldsymbol{i}}(\widetilde{\Psi}, \boldsymbol{V}^{n+1/2}, G^n, G^{n+1}, \delta t)$ in Eq. (7).

### 2.1.3 Spatial discretization

The discretization framework of IFS-FVM combines a structured-grid FD–FV method in the vertical with an unstructured[5] FV approach in the horizontal (Smolarkiewicz et al., 2016). The FV discretization and differentiation on the spherical surfaces adopt the median-dual approach described in Szmelter and Smolarkiewicz (2010). All dependent variables are co-located in the nodes in 3-D. The consistent spa-

---

[5]Note that although the presented IFS-FVM formulation assumes an unstructured mesh with indirect addressing in the horizontal, the model may exploit structured or semi-structured grids on future HPC architectures.

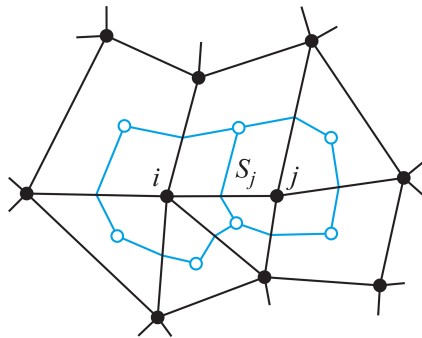

**Figure 1.** Schematic of the median-dual mesh in 2-D. The edge connecting nodes $i$ and $j$ of the primary polygonal mesh pierces, precisely in the edge centre, the face $S_j$ shared by computational dual cells surrounding nodes $i$ and $j$. Open circles represent geometrical barycentres of the primary mesh, solid black lines mark the primary mesh, and blue lines indicate dual cells with control volumes $\mathcal{V}_i$ and $\mathcal{V}_j$, respectively.

tial discretization of the applied MPDATA schemes in IFS-FVM, symbolized by the operator $\mathcal{A}_i$ in Eq. (7), is described in Kühnlein and Smolarkiewicz (2017).

The schematic in Fig. 1 illustrates an arbitrary unstructured mesh on a 2-D horizontal plane. The median-dual FV approach defines the control volume containing the node $i$ by connecting the barycentres of polygonal mesh cells encompassing the node $i$ with the midpoints of the edges originating in the node $i$. All geometric elements such as cell volume $\mathcal{V}_i$, cell face area $S_j$, and face normals $\boldsymbol{n}_j$ are evaluated from vector calculus in the computational space, i.e. in terms of $x$ and $y$ coordinates (see Sect. 2.1.1) on a zonally periodic horizontal plane (Szmelter and Smolarkiewicz, 2010). The unstructured FV discretization in the horizontal is combined with the standard second-order FD–FV method on the vertical structured grid with an independent coordinate $z$.

For a differentiable vector field $\boldsymbol{A}$, the Gauss divergence theorem $\int_\Omega \nabla \cdot \boldsymbol{A} = \int_{\partial\Omega} \boldsymbol{A} \cdot \boldsymbol{n}$ applied over the control volume surrounding node $\boldsymbol{i} = (k, i)$ in 3-D computational space is given as

$$(\nabla \cdot \boldsymbol{A})_{\boldsymbol{i}} = \frac{1}{\mathcal{V}_i} \sum_{j=1}^{l(i)} A_{k,j}^\perp S_j + \frac{A_{k+1/2,i}^z - A_{k-1/2,i}^z}{\delta z} , \qquad (22)$$

where $l(i)$ numbers the edges connecting node $(k, i)$ with its horizontal neighbours $(k, j)$, and $S_j$ refers both to the face per se and its surface area[6]. The geometric quantities $S_j$, $\mathcal{V}_i$, and $\delta z$ of the computational space are independent of height. The fields $A_{k,j}^\perp$ and $A_{k+1/2,i}^z$ are interpreted as the mean normal components of the vector $\boldsymbol{A}$ at the horizontal and vertical cell faces, respectively. Elementary approximations are

---

[6]Note that in IFS-FVM, $S_j$ and $\mathcal{V}_i$ have dimensions of length and area, and the actual face areas and volumes of prismatic cells are $S_j \delta z$ and $\mathcal{V}_i \delta z$, respectively, in computational space (Smolarkiewicz et al., 2016).

given as $A_{k,j}^\perp = 0.5\,\boldsymbol{n}_j \cdot (\boldsymbol{A}_{k,i} + \boldsymbol{A}_{k,j})$ in the horizontal and $A_{k+1/2,i}^z = 0.5(A_{k+1,i}^z + A_{k,i}^z)$ in the vertical (Szmelter and Smolarkiewicz, 2010; Smolarkiewicz et al., 2016). However, when applied in Eq. (22) without non-linear operations involved at the faces, cancellation of the node value $\boldsymbol{A}_{k,i}$ occurs, and this is exploited here to obtain a more efficient implementation of the median-dual approach by simply using $A_{k,j}^\perp = 0.5\,\boldsymbol{n}_j \cdot \boldsymbol{A}_{k,j}$ and $A_{k+1/2,i}^z = 0.5 A_{k+1,i}^z$. Similarly, applying the same cancellation of the node value $\Psi_{k,i}$ of a differentiable scalar field $\Psi$, the 3-D nabla operator $\nabla\Psi$ in computational space interpreted in terms of the Gauss divergence theorem is

$$(\nabla\Psi)_{k,i} = \left( \frac{1}{\mathcal{V}_i} \sum_{j=1}^{l(i)} 0.5\Psi_{k,j} S_j^x , \ \frac{1}{\mathcal{V}_i} \sum_{j=1}^{l(i)} 0.5\Psi_{k,j} S_j^y , \right.$$
$$\left. 0.5 \frac{\Psi_{k+1,i} - \Psi_{k-1,i}}{\delta z} \right), \qquad (23)$$

where $S_j^x$ and $S_j^y$ denote the $x$ and $y$ components of the oriented surface element $\boldsymbol{S}_j = S_j \boldsymbol{n}_j$. Given Eq. (22) and Eq. (23) in the computational space, they are augmented with metrics of the curvilinear coordinate framework to obtain the respective physical divergence and gradient appearing in the governing equations of Sect. 2.1.1 and 2.1.2, respectively. For illustration, one example for the revised implementation of the velocity divergence in the first term on the RHS of Eq. (17) at the node $(k, i)$ is

$$\left( \frac{1}{\mathcal{G}} \nabla \cdot (\mathcal{G} \widetilde{\mathbf{G}}^{\mathrm{T}} \boldsymbol{u}) \right)_{k,i} =$$
$$\frac{0.5}{\mathcal{G}_{k,i} \mathcal{V}_i} \sum_{j=1}^{l(i)} \mathcal{G}_{k,j} (\widetilde{G}_1^1 u)_{k,j} S_j^x + \mathcal{G}_{k,j} (\widetilde{G}_2^2 v)_{k,j} S_j^y$$
$$+ \frac{0.5}{\mathcal{G}_{k,i} \delta z} \left( \mathcal{G}_{k+1,i} \left( \widetilde{G}_1^3 u + \widetilde{G}_2^3 v + \widetilde{G}_3^3 w \right)_{k+1,i} \right.$$
$$\left. - \mathcal{G}_{k-1,i} \left( \widetilde{G}_1^3 u + \widetilde{G}_2^3 v + \widetilde{G}_3^3 w \right)_{k-1,i} \right). \qquad (24)$$

The details about the specification of the curvilinear coordinate framework under shallow- and deep-atmosphere equations are described in Appendix C.

For IFS-FVM, the mesh generation and mesh data structures, as well as the nearest-neighbour distributed-memory communication using MPI, are handled by ECMWF's Atlas library, comprehensively described in Deconinck et al. (2017). Atlas is also designed to make use of specific programming paradigms to support accelerators, although these have not yet been explored with IFS-FVM. For the quasi-uniform octahedral reduced Gaussian grid (Sect. 2.4), the parallelization of Atlas adopts the equal-region horizontal domain decomposition of IFS. The nearest-neighbour communications enabling the FV stencil operations in the parallel horizontal domain are performed on overlap regions (halos) between partitions, as well as a "peri-

odic overlap" for the east–west boundary on the spherical–spheroidal CE1 Earth. With the median-dual FV approach presented above, an overlap region of one element is typically used.

In the present work, the programming of the discrete differential operators (22) and (23) in the IFS-FVM modern Fortran code has been comprehensively revised from earlier implementations in Smolarkiewicz et al. (2016). While Smolarkiewicz et al. (2016) used a hybrid edge- and node-based programming already different from the edge-based codes described in Szmelter and Smolarkiewicz (2010), the current IFS-FVM programming represents a purely node-based implementation. When evaluating Eqs. (22) and (23), outer loops over nodes of the co-located grid (re)compute the required quantities on edges "on the fly". The code based on the resulting longer, fused node-based loops performs a larger overall number of computations as quantities on the edges are computed more than once, but it is overall more efficient as many intermediate memory stores are avoided and there is a greater chance for data in cache to be reused. Due to the underlying FD–FV discretization leading to stencil computations, IFS-FVM is naturally cache and memory-bandwidth bound. The node-based programming improves the flop-per-byte ratio and relaxes the dependency on relatively (compared to pure computation) slow memory access. As a consequence, it provides a significant gain in efficiency compared to the hybrid edge- and node-based programming applied in Smolarkiewicz et al. (2016). In addition, the fused loops and data locality in the node-based code effectively enabled the performance of the shared-memory parallelization with OpenMP, as the threads may operate on more local memory regions and avoid thread conflicts and cache trashing. An overall speed-up of IFS-FVM by a factor of $\sim$ 3–4 has been found on ECMWF's Cray XC40 by converting from the hybrid edge- and node-based to the purely node-based code.

We note that the GCR solver for the Helmholtz problem (20) uses global communications in the computation of inner products (characteristic of Krylov-subspace methods) and residual error norms. This has been implemented for the present paper such that summation is first done locally for each distributed-memory task, and the resulting single numbers are then reduced over all tasks.

## 2.2 Spectral-transform IFS

The spectral-transform IFS (denoted as IFS-ST in this paper) that is operational at ECMWF is based on the hydrostatic primitive equations (HPEs). The HPEs are formulated in a hybrid sigma–pressure terrain-following vertical coordinate following Simmons and Burridge (1981). The HPEs are integrated numerically using the efficient centred two-time-level SISL scheme, which permits long time steps due to the unconditional stability provided by the fully implicit treatment of the fast acoustic[7] and buoyant modes, and the 3-D SL advection (Ritchie et al., 1995; Temperton et al., 2001; Hortal, 2002). Therefore, the constant time step of the SISL integration in IFS-ST can be selected according to optimal efficacy rather than stability. The ST method, which is applied along model levels in the horizontal, is combined with a finite-element (FE) approach to discretize the integral operator in the vertical direction (Untch and Hortal, 2004)[8]. In 2002, this vertical FE scheme of Untch and Hortal (2004) with a co-located arrangement of prognostic variables replaced the former FD scheme of Simmons and Burridge (1981) with the Lorenz staggering. Prognostic variables of the HPEs and other main formulation features of IFS-ST are given in Table 1. Wedi et al. (2015) provide a recent overview of IFS-ST and a comprehensive list of references, while the official IFS documentation and changes with model cycles can be found on the ECMWF website (https://www.ecmwf.int/, last access: 6 February 2019).

The IFS-ST uses a discrete spherical harmonics representation of the spectral space (Wedi et al., 2013). At every time step, the model fields are transposed between grid-point, Fourier, and spherical harmonics representation. General concerns about the computational efficiency of the Legendre transforms between Fourier modes and spherical harmonics (Williamson, 2007) could be mitigated by adopting a fast Legendre transform (FLT; Wedi et al., 2013), which is employed together with fast Fourier transforms (FFTs). Furthermore, the increasing importance of non-linearities in the RHS forcing terms in the governing equations and aliasing at higher resolutions stimulated the adoption of a cubic truncation of the spherical harmonics in the ST method (Wedi, 2014). The cubic versus the former linear truncation basically samples the highest wavenumber with four instead of two grid points. Special treatment is required near the poles with the reduced grids for which it always approaches the linear truncation (Wedi, 2014). The extra sampling of a particular spectral resolution with a relatively larger grid size in the cubic truncation led to substantial further improvement in the efficiency and accuracy of the IFS-ST at ECMWF (Wedi et al., 2015). The cubic truncation in combination with the octahedral reduced Gaussian grid went operational at ECMWF in 2016. The nomenclature for this grid configuration is defined as "TCo" (for "triangular-cubic" truncation and "octahedral" grid) followed by the number of waves in spectral space (Malardel et al., 2016). The octahedral reduced Gaussian grid, which is also employed with IFS-FVM, is reviewed in Sect. 2.4.

The semi-Lagrangian advection scheme in IFS-ST is based on Ritchie et al. (1995). The SL trajectories are com-

---

[7]The HPEs analytically filter internal acoustic modes but support the external Lamb mode.

[8]The FE implementation of the discrete vertical integral operator is based on the Galerkin method using cubic B splines as basis functions.

puted with an iterative algorithm. At each iteration of the algorithm, the wind at the midpoint in time and space is re-evaluated using the second-order time-extrapolating algorithm SETTLS (stable extrapolation two-time-level scheme; Hortal, 2002). The two-time-level semi-implicit integration of IFS-ST follows the template

$$X_A^{n+1} = X_D^n + \mathcal{N}_M^{n+1/2} + \frac{1}{2}\left(\mathcal{L}_D^n + \mathcal{L}_A^{n+1}\right). \tag{25}$$

Here, $X$ represents the prognostic variables, the subscripts A, M, and D denote the arrival, middle, and departure points, respectively, and $n$ and $n+1$ again refer to the current and future time step. In addition, the operator $\mathcal{L}$ symbolizes the linear operator and $\mathcal{N}$ the non-linear residual $\mathcal{N} = \mathcal{M} - \mathcal{L}$, with $\mathcal{M}$ being the full non-linear model. The operator $\mathcal{L}$ results from the linearization of $\mathcal{M}$ with regard to a horizontally homogeneous reference state. Before the final semi-implicit solution with Eq. (25) at each time step, the explicit guess of the future model state $X$ is computed by replacing $\mathcal{L}_A^{n+1}$ with $\mathcal{L}_A^n$. Generally, $X_D^n$ is interpolated at the departure point by quasi-cubic interpolation. A horizontal quasi-monotonic interpolation is used for the horizontal wind, the temperature, and the surface pressure, and a 3-D quasi-monotonic limiter is used for the specific humidity. All the other water content variables are estimated at the departure point using linear interpolation and thus no monotonic filter is needed. The SL scheme in IFS-ST is applied to specific (per unit mass) variables whose equations are written in advective form. It is then intrinsically non-conservative (Malardel and Ricard, 2015), but global mass fixers have been developed for the atmospheric composition applications (Diamantakis and Flemming, 2014; Diamantakis and Augusti-Panareda, 2017). Notably, with the cubic truncation and the octahedral reduced Gaussian grid, global mass conservation is nearly exact in IFS; see Wedi et al. (2015).

The IFS also includes various research options which are not yet applied in the operational configuration, most notably the nonhydrostatic (NH) formulation based on the fully compressible equations (Bubnová et al., 1995; Bénard et al., 2010), which has been made available to ECMWF by Météo-France and the Aladin Consortium. The HPE and NH formulations of IFS-ST employ the same SISL integrators, but the NH extension requires a predictor–corrector approach – the so-called iterative-centred implicit (ICI) scheme (Bénard et al., 2010) – for stability in global configurations. The ICI scheme in IFS-ST requires recomputation of the semi-Lagrangian trajectory and interpolations in the corrector step, which is not needed in the predictor–corrector approach of IFS-FVM (Sect. 2.1.2). The prognostic variables and main characteristics of the NH formulation are also given in Table 1. Furthermore, the NH formulation is currently restricted to the vertical FD scheme by Bubnová et al. (1995). By default, the HPE and NH formulations of IFS-ST use the shallow-atmosphere approximation. In Sect. 3, we will com-

pare the computational performance of the HPE and NH formulations of IFS-ST against IFS-FVM.

## 2.3 Some aspects of physics–dynamics coupling

The IFS physics parameterization package at ECMWF is applied in the same configuration throughout the medium-range, sub-seasonal, and seasonal forecasting systems. The physics package includes parameterization of radiation, moist convection, clouds and stratiform precipitation, surface processes, sub-grid-scale turbulence, and orographic and non-orographic gravity wave drag.

In IFS-ST, the physics–dynamics coupling employs the SLAVEPP (semi-Lagrangian averaging of physical parameterization) scheme (Wedi, 1999). SLAVEPP targets second-order accuracy by averaging tendencies from selected physics parameterizations along the SL trajectory at the midpoint in space–time between the departure point at $t^n$ and arrival point[9] at $t^{n+1}$ CE2. Thereby, the tendencies at $t^{n+1}$ use a provisional first guess from the explicit dynamics as input, and the final tendencies are applied at the arrival point of the SL trajectory; see Wedi (1999) for details. The basic approach of the physics–dynamics coupling in the IFS uses sequential splitting of tendencies, i.e. the various processes are integrated one after another, and the updated tendencies are used as input to the subsequent process; see Beljaars et al. (2018) for discussion. More details about the IFS physics parameterizations can be found in the general IFS documentation.

The physics–dynamics coupling in IFS-FVM differs from IFS-ST. As explained in Sect. 2.1.2, the current implementation of IFS-FVM incorporates the tendencies from physics parameterizations $P^\Psi$ by means of a first-order coupling at $t^n$. Therefore, the fields that enter the parameterizations are from $t^n$, and there is no averaging between tendencies from $t^n$ and $t^{n+1}$. While the incorporation of the tendencies from physics parameterization deviates from IFS-ST, the sequential splitting between the various IFS physics parameterizations is kept exactly the same. Incorporating the physics parameterizations with first order at $t^n$ is motivated by the generally smaller time steps $\delta t$ in IFS-FVM than in IFS-ST and the desire to implement straightforward options for subcycling of the dynamics (see below). In addition, numerical experimentation with IFS-FVM so far has shown favourable results in terms of the incorporation of the physics at $t^n$. Nevertheless, different forms of coupling IFS-FVM to the physics parameterizations will be explored in the future.

The IFS-FVM code has its own interface to the IFS physics parameterizations. Among others, it involves conversion between IFS-FVM's variables and those employed

---

[9]SLAVEPP is applied to tendencies from radiation, moist convection, and the cloud scheme, whereas tendencies from turbulence and gravity wave drag parameterizations are incorporated with first order at $t^{n+1}$ (Wedi, 1999).

in IFS-ST (see Table 1)[10], interpolations to vertical interfaces, and the provision of local quantities describing the mesh geometry, but also a number of technical aspects due to some differences in the computational design of IFS-FVM and IFS-ST. Some of these operations in the interface may be removed at a later stage when the IFS physics parameterization package becomes more harmonized with IFS-FVM. However, generally the coupling is facilitated by common features of IFS-FVM and IFS-ST, such as longitude–latitude coordinates, the octahedral reduced Gaussian grid, and the co-located arrangement of dependent variables. The IFS-FVM interface to the physics parameterizations also includes an option for subcycling of the dynamics. The template scheme (7) for one physics time step from $t^N$ to $t^N + \Delta t_{\text{phys}} \equiv t^N + N_s \delta t$ can be written as $\ell = 1, N_s$:

$$
\Psi_i\left(t^N + \ell \delta t\right) = \mathcal{A}_i\big(\widetilde{\Psi}, \boldsymbol{V}(t^N + (\ell \delta t - 0.5)),
$$
$$
G(t^N + (\ell - 1)\delta t), G(t^N + \ell \delta t), \delta t\big)
$$
$$
+ b^\Psi \delta t \, \mathcal{R}_i^\Psi\left(t^N + \ell \delta t\right), \tag{26}
$$

where

$$
\widetilde{\Psi} = \Psi\left(t^N + (\ell - 1)\delta t\right) + a^\Psi \delta t \, \mathcal{R}^\Psi\left(t^N + (\ell - 1)\delta t\right)
$$
$$
+ \delta t \, P^\Psi\left(t^N, \Delta t_{\text{phys}}\right). \tag{27}
$$

The physics tendency $P^\Psi$ is evaluated with the physics time step $\Delta t_{\text{phys}}$ and is then reused for the $N_s$ subcycling steps with $\delta t$. The IFS-FVM has been coded rigorously with a variable time stepping capability (Kühnlein and Smolarkiewicz, 2017). In the case of the physics–dynamics coupling with subcycling Eq. (26), we adapt the semi-implicit time step $\delta t$ only every $N_s$ time steps with the corresponding physics time step given as $\Delta t_{\text{phys}} = N_s \delta t$. Apart from radiation[11], the current operational IFS-ST evaluates the physics parameterizations at every semi-implicit time step $\delta t$. However, due to the Eulerian versus SL advection, IFS-FVM uses considerably smaller time steps than IFS-ST[12], and therefore physics–dynamics coupling may use some form of subcycling as indicated above or other approaches, such as parallel splitting, to remain competitive. The cost per time step of the full IFS physics parameterization package can be up to 40 % of the forecast model. For the idealized DCMIP experiments considered in the present work, we focus on the parameterization of clouds and stratiform precipitation incorporated by means of the general IFS-FVM and IFS-ST interfaces described above.

---

[10]Examples are conversions between mixing ratios $r_k$ and specific water content variables $q_k$ or between quantities in the height-versus pressure-based coordinate systems.

[11]In the current high-resolution deterministic IFS forecasts on the O1280 grid, the radiation scheme is called every hour, compared to the semi-implicit model time step $\delta t = 450\,\text{s}$, and is also run on the coarser O400 grid.

[12]With the current formulations, the time step $\delta t$ in IFS-FVM is typically about a factor of 6–7 smaller than in IFS-ST.

## 2.4 Octahedral reduced Gaussian grid

As with the classical reduced Gaussian grid of Hortal and Simmons (1991), the octahedral reduced Gaussian grid (or simply "octahedral grid") (Wedi et al., 2015; Malardel et al., 2016; Smolarkiewicz et al., 2016) specifies the latitudes according to the roots of the Legendre polynomials. The two grids differ in the arrangement of the points along the latitudes, which follows a simple rule for the octahedral grid: starting with a minimum of $\sim 20$ points on the first latitude around the poles, four points are added with every latitude towards the equator, whereby the spacing between points along the individual latitudes is uniform and there are no points at the equator. The octahedral grid is suitable for transformations involving spherical harmonics. Figure 2 depicts the O24 octahedral grid nodes, together with the corresponding edges of the primary mesh as applied in IFS-FVM. Also shown is the dual mesh spacing of the O1280 grid. Compared to the classical reduced Gaussian grid of Hortal and Simmons (1991), the octahedral grid provides a much more uniform dual mesh resolution in the FV context (Malardel et al., 2016). Negligible grid imprinting in IFS-FVM with the octahedral grid will be shown by means of numerical experiments in Sect. 3.

## 3 Benchmark simulation results

We study the solution quality and also the computational efficiency of IFS-FVM in comparison to the established IFS-ST. The hydrostatic IFS-ST represents a proven formulation for global medium-range NWP at ECMWF, and the aim is to reproduce its results with the novel IFS-FVM.

Baroclinic instability represents a common and relevant test problem to evaluate the performance of global NWP models in the large-scale hydrostatic regime. The underlying processes are fundamental to the life cycle (i.e. formation to decay) of high- and low-pressure systems in the mid-latitude "storm tracks" of the Earth's atmosphere. Here, we adopt the experimental set-up for a baroclinic wave life cycle used in the 2016 edition of DCMIP following Ullrich et al. (2014) and the documentation available in Ullrich et al. (2016). Note that the IFS is a highly optimized single-application model for real-time numerical weather prediction, and adapting the code to idealized configurations is not straightforward. Because of this and also a particular interest in requirements and applications at ECMWF, we consider configurations and physics parameterizations that depart from the test case specifications of DCMIP in Ullrich et al. (2016).

To study the accuracy of the novel nonhydrostatic IFS-FVM based on the finite-volume discretization, we verify its solution quality against IFS-ST based on the spectral-transform approach. This comparison is performed in Sect. 3.1 for dry adiabatic simulations of the baroclinic instability, i.e. dynamical core only, and then in Sect. 3.2 un-

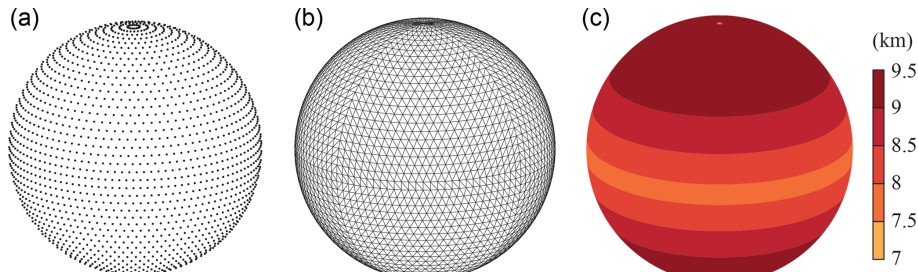

**Figure 2.** The locations of the octahedral reduced Gaussian grid nodes are shown in **(a)**, using for illustration the very coarse O24 grid example with 24 latitudes between the pole and equator. Panel **(b)** depicts the associated edges of the primary mesh connecting the nodes as applied in the context of the FV discretization of IFS-FVM. Panel **(c)** provides the local spacing of the FV dual mesh for the O1280 grid corresponding to the highest-resolution deterministic forecast model currently at ECMWF.

der consideration of moist-precipitating processes that involve coupling to the prognostic single-moment bulk microphysics parameterization of the operational IFS at ECMWF (Forbes et al., 2010). The computational efficiency of IFS-FVM alongside the hydrostatic and nonhydrostatic IFS-ST is studied in Sect. 3.3. Note that we use the nonhydrostatic IFS-ST only when looking at the efficiency of the various dynamical core formulations. In terms of the solution quality for the baroclinic instability, nonhydrostatic effects are entirely negligible at the considered coarse resolutions (see next paragraph), and therefore only results with the hydrostatic IFS-ST are shown and analysed.

We use two different sizes of the octahedral reduced Gaussian grid for comparing the solution quality of IFS-FVM and IFS-ST. Considered are very coarse (O160,TCo159) and coarse (O320,TCo319) grids by current NWP standards, corresponding to about 64 and 32 km nominal horizontal grid spacings, respectively; see Sect. 2.2 and 2.4 for the nomenclature that defines the grids. For the two horizontal grid sizes, both IFS-FVM and IFS-ST employ 60 stretched vertical levels. The height coordinate of IFS-FVM (Appendix C) is specified exactly according to the computed height of the hybrid sigma–pressure coordinate of IFS-ST at the initial time of the simulation. The lowest full level is located at a height of about 10 m, and the vertical spacing between model levels ranges from about 12 m near the ground to 4 km near the model top located at 48 km. In contrast to the height coordinate levels in IFS-FVM, the hybrid sigma–pressure coordinate levels in IFS-ST change with time, but overall the vertical spacing remains quite similar in both models over the course of the 15-day simulations. The similar spacing is particularly true for the vertical levels near the surface, where the terrain-following character of the hybrid sigma–pressure coordinate dominates. Note that, as the maximum local difference in height of the lowest full levels is found to be smaller than 1 m over the 15-day baroclinic instability simulation, output fields of IFS-FVM and IFS-ST may be straightforwardly compared at this level without using inter-

polation[13]. For the comparison of computational efficiency in Sect. 3.3, we employ the (O1280,TCo1279) horizontal grid, corresponding to about 9 km spacing, with 137 stretched vertical levels, again with a similar distribution in IFS-FVM and IFS-ST. With regard to the time step size, IFS-ST uses constant increments $\delta t$ of 1800, 1200, and 450 s for the TCo159, TCo319, and TCo1279 grids, respectively. In contrast, IFS-FVM generally applies variable time stepping that targets a maximum horizontal advective Courant number of 0.95 – the actual maximum 3-D Courant number can be significantly larger than 1 as this is permitted by the horizontal–vertical split NFT advection in IFS-FVM (Appendix A). As explained in Sect. 2.1.2, we have implemented the integration of Eq. (17) using an off-centred variant of the template semi-implicit scheme (7). All IFS-FVM results presented in this paper used the backward Euler scheme $\alpha = 1$, as this has been the default so far in previous implementations. However, one exception to this is the middle panel in Fig. 5, which shows, for comparison to the default $\alpha = 1$, the corresponding result obtained with the weakly off-centred trapezoidal scheme using $\alpha = 0.51$.

All IFS-FVM and IFS-ST results presented in this section were obtained without any explicit diffusion or regularization.

For the dry and moist configurations, the baroclinic instability evolution starts from two zonal jet flows in the midlatitudes of each global hemisphere that are in thermal wind balance with the meridional temperature gradient. The definition of the balanced initial state is given by analytical functions provided in Ullrich et al. (2014). Here, this balanced zonal flow is also used to specify the ambient state variables $\boldsymbol{u}_a(y, z)$, $\theta_a(y, z)$, $\pi_a(y, z)$ of the governing fully compressible Eqs. (1a)–(1e) that fulfil Eq. (4). A local zonal velocity perturbation in the form of a simple exponential bell (tapered to zero in the vertical) excites the instability, leading to eastward-propagating Rossby modes (Ullrich et al., 2016).

---

[13]For instance, the 1 m height difference corresponds to about 0.1 hPa near the surface, which is negligible with regard to the subsequent analysis.

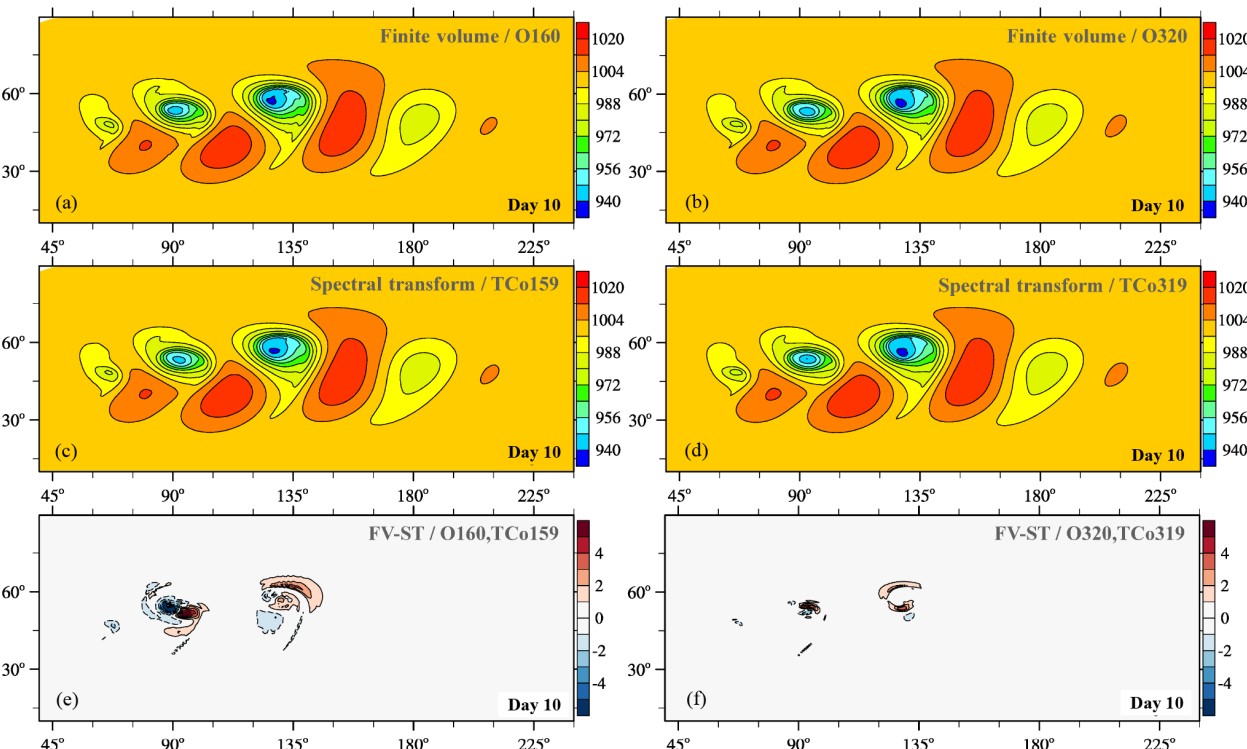

**Figure 3.** Dry baroclinic instability at day 10: panels **(a)**–**(d)** show pressure on the lowest full level (hPa) obtained with IFS-FVM and IFS-ST, while **(e)** and **(f)** depict the corresponding difference between the solutions. Panels **(a)**, **(c)**, **(e)** and **(b)**, **(d)**, **(f)** are for the (O160,TCo159) and (O320,TCo319) horizontal grids, respectively. TS8

Here, we apply the triggering zonal velocity perturbation in both the northern and southern global hemispheres. This dual triggering departs from Ullrich et al. (2016), in which the perturbation was only applied in the Northern Hemisphere, but it permits a clean evaluation of kinetic energy spectra relatively early in the baroclinic wave evolution, enables the study of the solution symmetry about the equator, and is more relevant to real weather. Nevertheless, for reference we provide one illustration in Fig. 6 for the set-up in which the triggering of the baroclinic instability is applied in the Northern Hemisphere only. After an initial period of linear growth, the instability enters the non-linear stage from 6–7 days of simulation time. Our analysis will focus on simulation results at day 10 and day 15.

### 3.1 Results for dry simulations

Figures 3 and 4 present the horizontal cross section of near-surface pressure and the zonal height cross section of meridional wind, respectively, for the dry adiabatic simulations at day 10. At this stage, a large-amplitude baroclinic wave has developed and formed sharp fronts in the lower troposphere. Generally very close agreement is found between the finite-volume (IFS-FVM) and spectral-transform (IFS-ST) solutions. This is emphasized by the difference plots in the bottom row of the horizontal and vertical cross sec-

tions in Figs. 3 and 4, respectively. The solutions show identical phase propagation and amplitude of the baroclinic wave throughout the entire vertical depth of the simulation domain, which applies to the very coarse (O160,TCo159) and coarse (O320,TCo319) grids. Where present, differences between IFS-FVM and IFS-ST become smaller with the higher resolution. Figure 5 compares the pressure field much later into the non-linear baroclinic instability evolution at day 15 for the finer of the two grids. This depiction shows close agreement between IFS-FVM (using the default $\alpha = 1$ as well as the $\alpha = 0.51$) and IFS-ST also at this later stage. The various dynamical core formulations provide visibly symmetric solutions around the equator, and there are no signs of any significant grid imprinting at the scale of the wavenumber-four irregularities of the octahedral grid. The latter is corroborated by Fig. 6 that shows the analogous result to Fig. 5 (again only $\alpha = 1$ for IFS-FVM), but in contrast to all other plots, the trigger of the baroclinic instability is applied in the Northern Hemisphere only.

Kinetic energy spectra evaluated at day 15 in Fig. 7 reveal a strikingly similar distribution of variance across wavenumbers from 1 to $\sim 200$. As all the simulations are without any explicit diffusion, the IFS-FVM spectra at the high wavenumbers attest to the implicit scale-selective regularization with artificial viscosity provided by the non-oscillatory

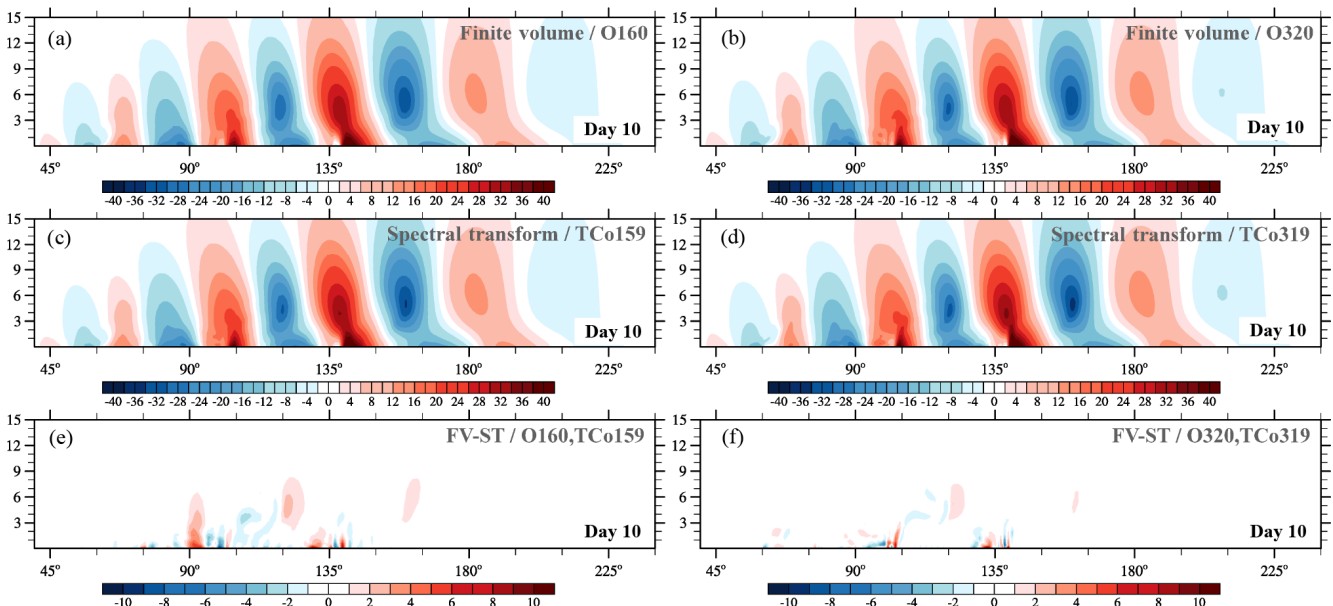

**Figure 4.** Dry baroclinic instability at day 10: panels **(a)**–**(d)** show meridional wind $v$ (m s$^{-1}$) along a zonal height cross section at 50° N obtained with IFS-FVM and IFS-ST, while **(e)** and **(f)** depict the difference of $v$ (m s$^{-1}$) between their solutions. Panels **(a)**, **(c)**, **(e)** and **(b)**, **(d)**, **(f)** are for the (O160,TCo159) and (O320,TCo319) horizontal grids, respectively.

finite-volume MPDATA advection[14]. The slope of the IFS-ST and IFS-FVM spectra with respect to wavenumber $l$ is somewhat shallower than $l^{-3}$ at large scales. This is consistent with results from other dynamical cores for this baroclinic instability test case studied in the context of the High-Impact Weather Prediction Project (HIWPP; Whitaker, 2014). The spectra of IFS-ST feature some accumulation of energy near the scale of the triangular-cubic truncation, corresponding to 4 times the grid spacing. This increased energy at these scales does not grow and is to a large extent controlled by the spectral filtering of the non-linear terms at every time step, among other mechanisms of implicit dissipation such as the semi-Lagrangian interpolation.

### 3.2 Simulation results for moist-precipitating configuration with the IFS cloud parameterization

Next we present results for the moist-precipitating baroclinic instability with coupling to the IFS cloud parameterization. Figure 8 shows the instantaneous large-scale precipitation rate at the surface[15] for the (O160,TCo159) and (O320,TCo319) grids at day 10. For any of these grids, both model formulations show five rainbands with essentially identical phase, as emphasized by the overlay with the 0.5 mm day$^{-1}$ black contour line of the corresponding

---

[14]The unsplit NFT MPDATA advection (Kühnlein and Smolarkiewicz, 2017) features lower implicit diffusion near the grid scale than the split advection applied here.

[15]The precipitation rate represents the liquid and rain (excluding ice and snow) sedimentation flux at the surface.

other model formulation. The elongated rainbands are associated with the lifting along sharp frontal zones. Precipitation amounts are overall similar but somewhat higher local values exist for IFS-FVM, particularly in the two easternmost rainbands when looking at the (O160,TCo159) grid. Figure 9 is analogous to Fig. 8 but for day 15. As can be expected, the spread between the different model formulations becomes larger. However, there is still reasonably close agreement, especially for the higher-resolution grid (O320,TCo319) in the right column of Fig. 9. Here, the locations of the easternmost frontal zone and associated rainband agree closely considering the late stage of the baroclinic instability evolution. Figure 10 supplements the precipitation plots with the corresponding pressure field on day 15. In addition to the standard configurations of IFS-FVM and IFS-ST, for which the physics parameterization is evaluated every dynamic time step $N_s = 1$, Fig. 10 also provides the IFS-FVM result with subcyling (middle panel) whereby the parameterizations are evaluated every $N_s = 3$ semi-implicit time step $\delta t$; see Sect. 2.3 for a discussion of the physics–dynamics coupling. Again, the pressure fields of all three simulations resemble each other closely, often even in the location and magnitude of smaller structures, while the modified physics–dynamics coupling frequency $N_s = 3$ to the cloud parameterization seems to have only a small impact on the solution. Furthermore, none of the simulations show significant grid imprinting in the pressure fields, but the solution symmetry about the equator is broken in both IFS-FVM and IFS-ST as a result of the incorporation of the cloud parameterization (in contrast to the dry results shown before in Fig. 5). The anal-

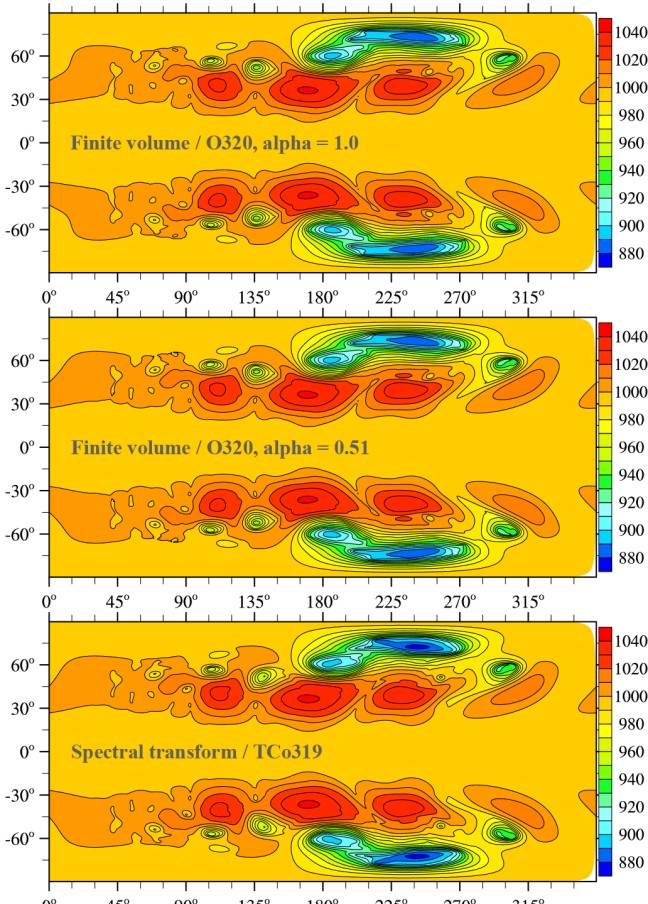

**Figure 5.** Dry baroclinic instability at day 15: pressure on the lowest full level (hPa) obtained with IFS-FVM (default backward Euler scheme with $\alpha = 1$ or trapezoidal scheme with weak off-centring $\alpha = 0.51$ in the integration of Eq. (17) for the Exner pressure perturbation $\varphi'$) and IFS-ST using the (O320,TCo319) grid.

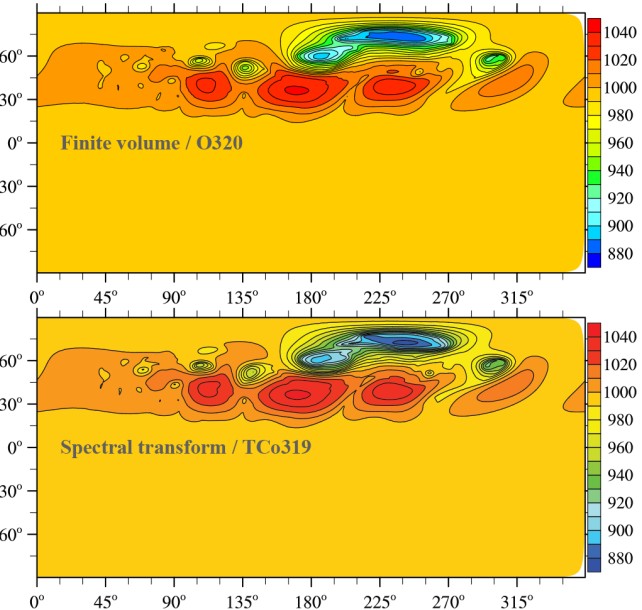

**Figure 6.** Dry baroclinic instability at day 15 when the triggering of the baroclinic instability was applied in the Northern Hemisphere only: pressure on the lowest full level (hPa) obtained with IFS-FVM and IFS-ST using the (O320,TCo319) grid.

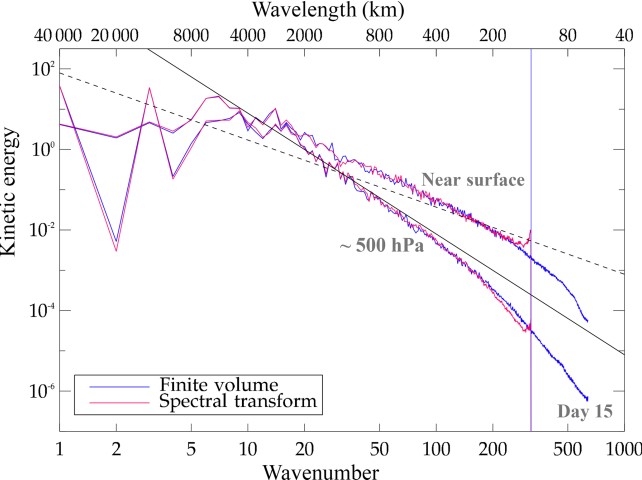

**Figure 7.** Dry baroclinic instability at day 15: kinetic energy spectra obtained with IFS-FVM and IFS-ST using the (O320,TCo319) grid. The blue vertical line indicates the spatial scale corresponding to 4 times the nominal grid spacing of the O320 octahedral grid, which also represents the cubic truncation scale with TCo319 applied in IFS-ST. The spectra are shown on model levels near the surface and at $\sim 500$ hPa.

ysis of the simulations is supplemented in Fig. 11 with the time series of the minimum near-surface pressure (panel a) and the area-integrated precipitation rate (panel b). The temporal evolution of these two quantities is close between IFS-FVM and IFS-ST. Particularly, the minimum near-surface pressure agrees almost exactly at day 15, although small differences occur over the course of the simulation. The onset and subsequent increase in precipitation matches well in IFS-FVM and IFS-ST, and the later variations in the precipitation rate are similar, with no systematic underestimation or overestimation. Kinetic energy spectra evaluated at day 15 are shown in Fig. 12. Compared to the spectra of the dry simulations in Fig. 7, IFS-FVM and IFS-ST consistently show a considerably larger kinetic energy in the scales smaller than wavenumber $\approx 120$ in the mid-troposphere at about 500 hPa. Overall, the presented consistent results of IFS-FVM and IFS-ST attest to the quality of the presented dry and moist-precipitating FV formulations along with the coupling to the IFS physics parameterization.

## 3.3 Computational efficiency

The computational efficiency of NWP models is crucial. For current HPC architectures and model resolutions, the operational IFS-ST at ECMWF represents one of the most efficient dynamical core formulations for global NWP. The IFS-

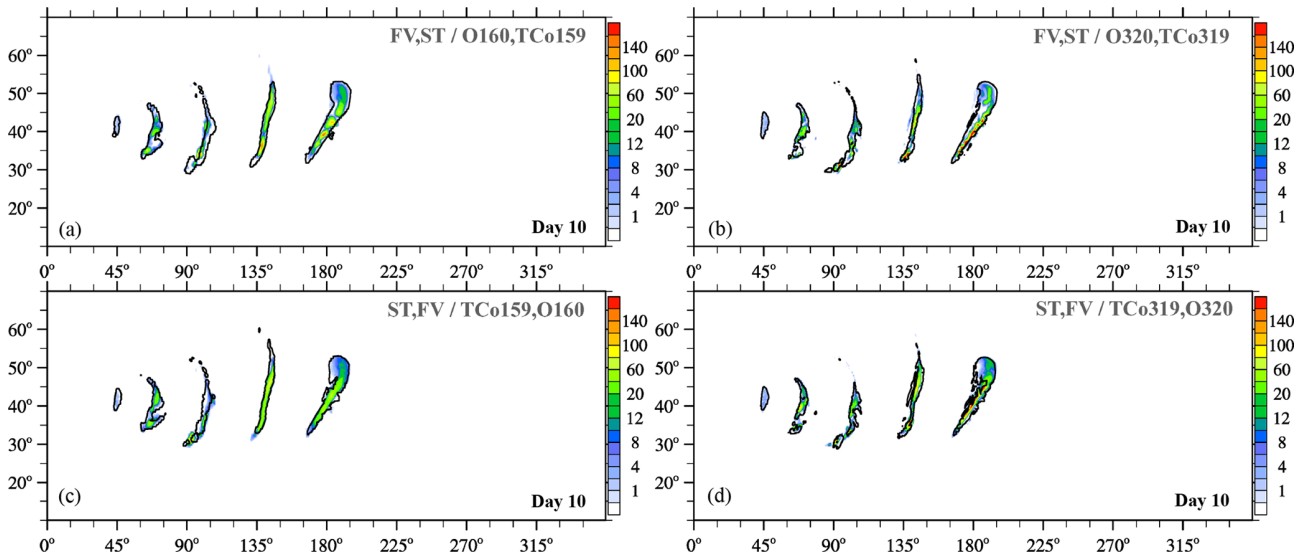

**Figure 8.** Moist-precipitating baroclinic instability at day 10: surface precipitation rate (mm day$^{-1}$) obtained with IFS-FVM and IFS-ST coupled to the same IFS cloud microphysics parameterization. The upper (lower) panels show shaded contours from the IFS-FVM (IFS-ST) simulations, overlaid by the IFS-ST (IFS-FVM) black contour line of 0.5 mm day$^{-1}$. Panels **(a)**, **(c)** and **(b)**, **(d)** are for the (O160,TCo159) and (O320,TCo319) horizontal grids, respectively.

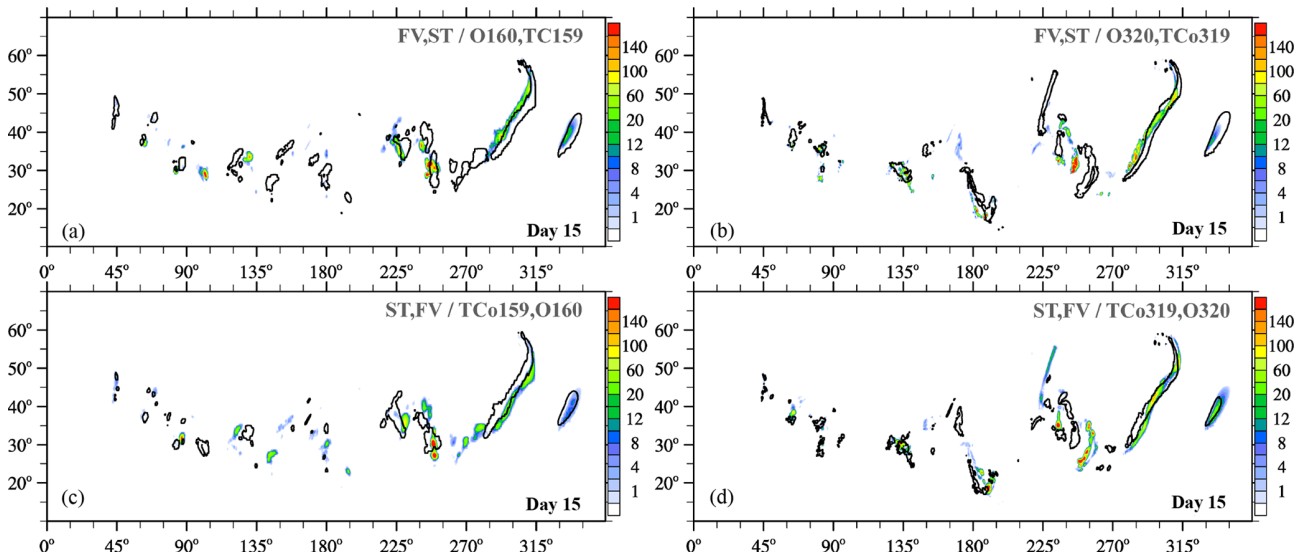

**Figure 9.** Moist-precipitating baroclinic instability at day 15: surface precipitation rate (mm day$^{-1}$) obtained with IFS-FVM and IFS-ST coupled to the same IFS cloud microphysics parameterization. The upper (lower) panels show shaded contours from the IFS-FVM (IFS-ST) simulations, overlaid by the IFS-ST (IFS-FVM) black contour line of 0.5 mm day$^{-1}$. Panels **(a)**, **(c)** and **(b)**, **(d)** are for the (O160,TCo159) and (O320,TCo319) horizontal grids, respectively.

FVM is envisaged for future applications in the nonhydrostatic regime running on future HPC architectures, but its computational performance on the current HPC facility at ECMWF sheds some light on its potential; see also Kühnlein and Smolarkiewicz (2019). Of interest is the relative performance of IFS-FVM to both the hydrostatic IFS-ST and its nonhydrostatic extension (see Sect. 2.2). In order to emphasize elementary aspects of the dynamical cores, here we assess the efficiency of the dry formulations only; i.e. no tracers or moisture variables, no physics parameterizations or coupled models, no I/O, and minimal diagnostics. Furthermore, we use the baroclinic instability benchmark of Sect. 3.1 but configured similar to HRES–ECMWF's highest-resolution deterministic forecast model.

Figure 13 highlights the runtimes of the three different dynamical core formulations. The HRES forecast

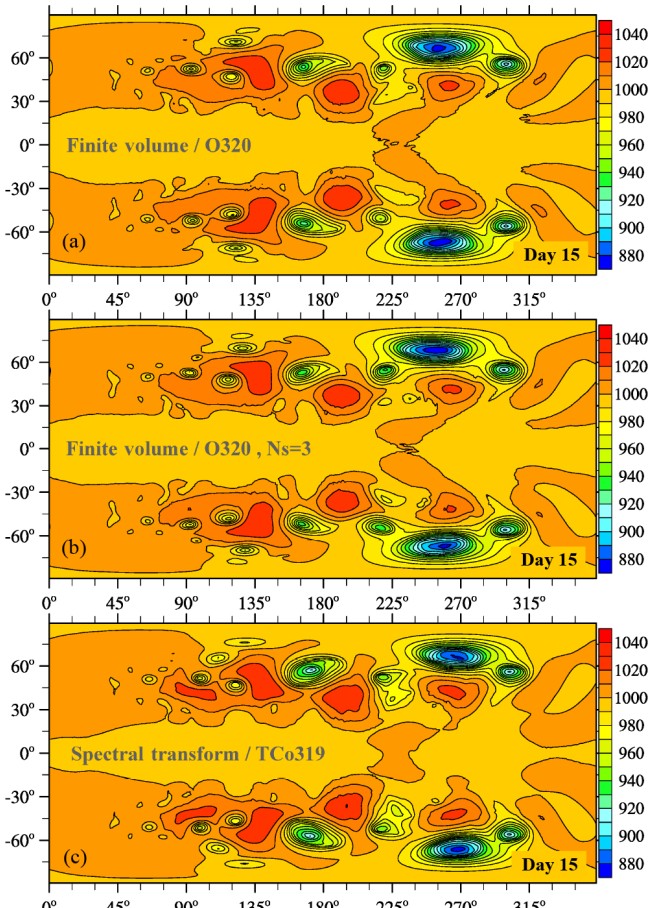

**Figure 10.** Moist-precipitating baroclinic instability at day 15: pressure on the lowest full level (hPa) obtained with IFS-FVM and IFS-ST coupled to the same IFS cloud microphysics parameterization. The IFS-FVM results in **(a)** and **(b)** employed the standard coupling at every time step $N_s = 1$ or subcycling of dynamics for three time steps $N_s = 3$, respectively. The simulations were performed with the (O320,TCo319) grid.

model configuration at ECMWF is currently based on the O1280/TCo1279 horizontal grid (corresponding to about 9 km grid spacing; Fig. 2c) with 137 vertical levels and run using 350 nodes (about one electrical group) on ECMWF's Cray XC40 supercomputer[16]. Importantly, while IFS-ST used the constant time step of 450 s, IFS-FVM employed variable time stepping according to the maximum permitted advective Courant number; therefore, in order to obtain realistic numbers for IFS-FVM, the timings were evaluated between day 10 and 15 in the fully non-linear stage of the baroclinic instability evolution.

---

[16]Each node on this supercomputer consists of two Intel Xeon EP E5-2695 v4 "Broadwell" processors, each with 18 cores, which for the 350 compute nodes employed results in a total of 12 600 cores. Here, a hybrid MPI–OpenMP parallelization with six threads was used by all three dynamical cores.

Figure 13 reveals that the time to solution that can be achieved with IFS-FVM for this configuration is only about twice as large as the operational hydrostatic IFS-ST and compares favourably with the nonhydrostatic IFS-ST. Although these performance measures merely represent snapshots at the current state of development, they highlight the potential of the numerical integration schemes applied in IFS-FVM to become competitive with state-of-the-art operational global weather forecasting models. Important aspects are that the FV method offers the prospect of better scalability and efficiency with respect to future HPC and that IFS-FVM employs substantially smaller time steps (again, about a factor 6–7 smaller compared to IFS-ST), which can be beneficial for accuracy. Further significant efficiency improvements of the IFS-FVM dynamical core are in preparation, and the work will be extended to physics–dynamics coupling with the smaller time steps.

## 4 Conclusions

Supporting substantially higher resolution in global NWP may ultimately demand local numerical discretizations to solve the governing nonhydrostatic equations in NWP models in a computationally efficient manner. The IFS-FVM successfully implements such a discretization and thus complements the operational hydrostatic IFS-ST and its nonhydrostatic extension at ECMWF. At the same time, the IFS-FVM introduces several useful new features into the IFS, such as conservative and monotone advective transport, deep-atmosphere all-scale governing equations, and fully flexible unstructured FV meshes with optional variable resolution or meshes defined about the nodes of the operational octahedral grid.

The paper highlighted the semi-implicit NFT finite-volume integration of the fully compressible equations of the novel IFS-FVM considering comprehensive moist-precipitating dynamics with coupling to the IFS cloud parameterization by means of a generic interface applicable for coupling to the full IFS physics parameterization package. Developments such as the new horizontal–vertical directionally split NFT advective transport scheme based on MP-DATA, variable time stepping, effective preconditioning of the Krylov-subspace solver for the elliptic Helmholtz problem arising in the semi-implicit scheme, and an efficient node-based implementation of the median-dual FV approach provide a basis for the overall efficacy of IFS-FVM and application in global NWP at ECMWF.

The IFS-ST is applied successfully for operational forecasting at ECMWF and is therefore considered an appropriate reference model. It was shown that the presented semi-implicit NFT finite-volume integration scheme on co-located meshes can achieve comparable solutions to the proven spectral-transform IFS-ST. Here, the study focused on medium- and extended-range simulation of the dry and

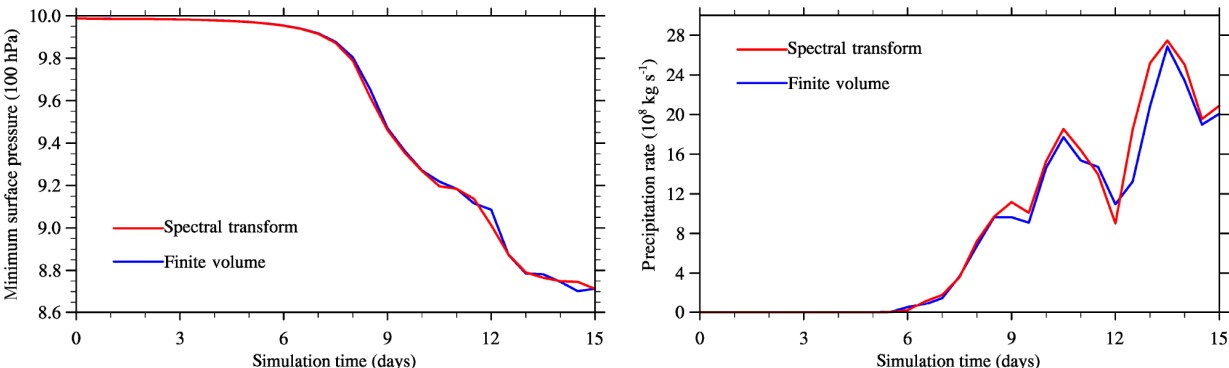

**Figure 11.** Moist-precipitating baroclinic instability: time series of minimum pressure on the lowest full level **(a)** and area-integrated rain rate **(b)**. The blue and red lines correspond to the IFS-FVM and IFS-ST results, respectively.

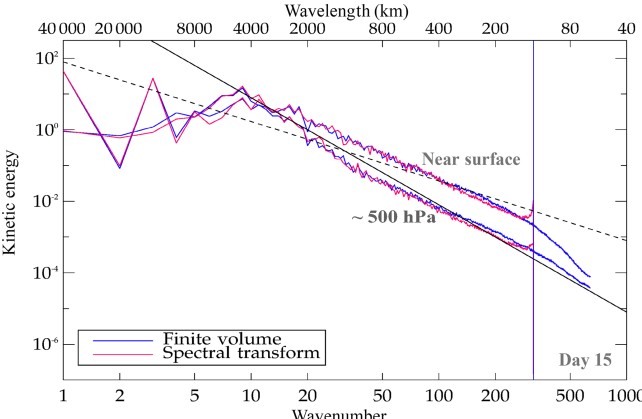

**Figure 12.** Moist-precipitating baroclinic instability at day 15: kinetic energy spectra obtained with IFS-FVM and IFS-ST using the (O320,TCo319) grid. The blue vertical line indicates the spatial scale corresponding to 4 times the nominal grid spacing of the O320 octahedral grid, which also represents the cubic truncation scale with TCo319 applied in IFS-ST. The spectra are shown on model levels near the surface and at ∼ 500 hPa.

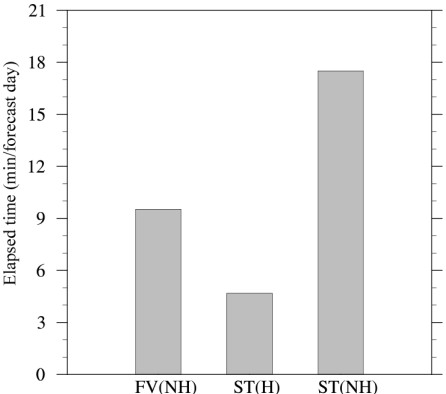

**Figure 13.** Elapsed time to run 1 day of the dry baroclinic instability benchmark similar to the current HRES configuration at ECMWF, i.e. the three different models – the nonhydrostatic IFS-FVM designated as FV(NH), the hydrostatic IFS-ST designated as ST(H), and the nonhydrostatic IFS-ST designated as ST(NH) – are set up for the O1280/TCo1279 horizontal grid (corresponding to about 9 km grid spacing) with 137 stretched vertical levels and employ 350 nodes of ECMWF's Cray XC40.

moist-precipitating baroclinic instability benchmark at various resolutions. While the baroclinic instability benchmark aims at global atmospheric dynamics in the hydrostatic regime, referenced supplementary studies with IFS-FVM emphasize non-orographic and orographic flows in the nonhydrostatic regime. In addition to solution quality, we have demonstrated highly competitive computational efficiency of the presented semi-implicit NFT finite-volume integration of IFS-FVM in comparison to the semi-implicit semi-Lagrangian integration of IFS-ST.

Common aspects of the finite-volume and spectral-transform model formulations are the octahedral reduced Gaussian grid, the co-location of variables, the geospherical framework, and the physics parameterizations. Sharing these properties facilitates the comparison of the different discretizations and physics–dynamics coupling. Moreover, it

provides numerous benefits for the general IFS model infrastructure, data assimilation, and ensemble system. Ongoing work advances IFS-FVM to full-physics global medium-range NWP at convection-resolving resolutions (Kühnlein and Smolarkiewicz, 2019).

*Code availability.* Model codes developed at ECMWF are the intellectual property of ECMWF and its member states, and therefore the IFS code is not publicly available. Access to a reduced version of the IFS code may be obtained from ECMWF under an OpenIFS licence (see http://www.ecmwf.int/en/research/projects/openifs for further information, last access: 6 February 2019).

*Data availability.* The model output data can be downloaded from https://doi.org/10.5281/zenodo.1445597 (Kühnlein et al., 2018).

## Appendix A: Horizontal–vertical splitting of the NFT advective transport

We consider the advection operator $\mathcal{A}_i$ in the two-time-level semi-implicit integration scheme (7) to be directionally split in the horizontal and vertical directions. This splitting is motivated by the observation that NWP models typically have a larger restriction on the time step in the vertical than the horizontal direction. For example, in the current operational configuration of the IFS run at TCo1279/L137 ($\approx 9$ km horizontal grid spacing and 137 stretched vertical levels), the advective Courant numbers are up to a factor of 2 larger in the vertical than in the horizontal direction. The horizontal–vertical splitting also accommodates IFS-FVM's unstructured horizontal discretization, enabling broad classes of meshes over the surface of the Earth's sphere–spheroid CE3, and the structured grid in the (stiff) vertical direction.

The proposed scheme implements mass-compatible second-order Strang splitting as explained in the following. The overall semi-implicit integration of the fully compressible Eqs. (1a)–(1e) proceeds exactly as explained in Sect. 2.1.2, but with the 3-D NFT advection operator $\mathcal{A}_i$ split into purely horizontal $\mathcal{A}_i^{xy}$ and vertical $\mathcal{A}_i^{z}$ schemes, respectively. For each model time step $\delta t$, these are applied in the sequence $\mathcal{A}_i^{z} \rightarrow \mathcal{A}_i^{xy} \rightarrow \mathcal{A}_i^{z}$ using half-time steps in the two vertical sweeps and the full time step in the horizontal part. Specifically, the split scheme commences with the integration of the mass continuity Eq. (1a) as

$$\rho_{\mathrm{d}i}^{[1]} = \mathcal{A}_i^{z}(\rho_{\mathrm{d}}^n, (v^z \mathcal{G})^{n+1/2}, \mathcal{G}^n, \mathcal{G}^{[1]}, 0.5\delta t) ,$$
$$\rho_{\mathrm{d}i}^{[2]} = \mathcal{A}_i^{xy}(\rho_{\mathrm{d}}^{[1]}, (v_{\mathrm{h}} \mathcal{G})^{n+1/2}, \mathcal{G}^{[1]}, \mathcal{G}^{[2]}, \delta t) ,$$
$$\rho_{di}^{n+1} = \rho_{\mathrm{d}i}^{[3]} = \mathcal{A}_i^{z}(\rho_{\mathrm{d}}^{[2]}, (v^z \mathcal{G})^{n+1/2}, \mathcal{G}^{[2]}, \mathcal{G}^{n+1}, 0.5\delta t) , \quad \text{(A1)}$$

which provides the updated densities $\rho_{\mathrm{d}}^{[1]}$, $\rho_{\mathrm{d}}^{[2]}$, $\rho_{\mathrm{d}}^{n+1}$ and accumulates normal mass fluxes $(v^z \mathcal{G} \rho_{\mathrm{d}})^{[1]}$, $(v_{\mathrm{h}}^{\perp} \mathcal{G} \rho_{\mathrm{d}})^{[2]}$, $(v^z \mathcal{G} \rho_{\mathrm{d}})^{[3]}$ for the three sub-steps. For compatibility with mass continuity, these quantities are then all employed in the subsequent advective transport of scalar variables $\widetilde{\Psi}$ (Eq. 8) as TS9

$$\Psi_i^{[1]} = \mathcal{A}_i^{z}(\widetilde{\Psi}, (v^z \mathcal{G} \rho_{\mathrm{d}})^{[1]}, (\mathcal{G} \rho_{\mathrm{d}})^n, (\mathcal{G} \rho_{\mathrm{d}})^{[1]}, 0.5\delta t) ,$$
$$\Psi_i^{[2]} = \mathcal{A}_i^{xy}(\Psi^{[1]}, (v_{\mathrm{h}}^{\perp} \mathcal{G} \rho_{\mathrm{d}})^{[2]}, (\mathcal{G} \rho_{\mathrm{d}})^{[1]}, (\mathcal{G} \rho_{\mathrm{d}})^{[2]}, \delta t) ,$$
$$\widehat{\Psi}_i = \Psi_i^{[3]} = \mathcal{A}_i^{z}(\Psi^{[2]}, (v^z \mathcal{G} \rho_{\mathrm{d}})^{[3]}, (\mathcal{G} \rho_{\mathrm{d}})^{[2]},$$
$$(\mathcal{G} \rho_{\mathrm{d}})^{n+1}, 0.5\delta t) . \quad \text{(A2)}$$

In Eqs. (A1) and (A2), the implementation of the horizontal advection transport $\mathcal{A}^{xy}$ follows the horizontal part of the unstructured-mesh FV MPDATA of Kühnlein and Smolarkiewicz (2017). The vertical scheme $\mathcal{A}^z$ is a corresponding 1-D structured-grid MPDATA. Results from numerical experimentation relevant to NWP show that the presented horizontally–vertically split NFT scheme based on MPDATA can be considerably more efficient than the standard fully

multidimensional (unsplit) MPDATA of Kühnlein and Smolarkiewicz (2017). This is particularly due to the integration of the vertical parts $\mathcal{A}_i^z$ with $\delta t/2$ each, which mitigates the vertical stability restriction while not adding any significant computational cost[17]. Overall, the horizontal–vertical splitting of $\mathcal{A}_i$ can enable a more than twice larger time step in the integration than the unsplit formulation. In addition, the split scheme facilitates the application of higher-order, e.g. Waruszewski et al. (2018), and/or flux-form semi-Lagrangian advective transport in the vertical. While a detailed presentation and analysis will be provided in a future publication, results so far indicate a comparable solution quality of the split versus unsplit schemes for global atmospheric flow benchmarks. All IFS-FVM results presented in this paper were obtained using the split scheme (A1)–(A2) for $\mathcal{A}_i$ in Eqs. (7)–(8).

## Appendix B: Weighted line Jacobi preconditioner

The bespoke preconditioner solves for the solution error $e$ of the pressure perturbation variable $\varphi'$:

$$\mathcal{P}(e) = \hat{r}, \quad \text{(B1)}$$

where $\hat{r}$ denotes the residual error of Eq. (20). The preconditioning operator $\mathcal{P}$ is then decomposed into vertical and horizontal parts (Smolarkiewicz and Margolin, 2000), and the residual problem is solved iteratively according to

$$\mathcal{P}_z(e^{\mu+1}) + \mathcal{P}_{\mathrm{h}}(e^{\mu}) - \hat{r} = 0, \quad \text{(B2)}$$

where $\mu$ numbers the iterations, of which there are typically two. The vertical part $\mathcal{P}_z$ is inverted directly with a tridiagonal algorithm. The horizontal part $\mathcal{P}_{\mathrm{h}}$ is lagged behind, except for its diagonal entries. The actual implementation is given as

$$\mathcal{P}_z(e^{\mu+1}) + \mathcal{P}_{\mathrm{h}}(e^{\mu}) + \mathcal{D}(e^{\mu+1} - e^{\mu}) - \hat{r} = 0 , \quad \text{(B3)}$$

where $\mathcal{D}$ is the diagonal coefficient of $\mathcal{P}_{\mathrm{h}}$, specified as

$$\mathcal{D}_{k,i} = -\frac{1}{4\mathcal{V}_i} \sum_{\ell=1}^{3} \frac{A_{\ell,i}^{\star}}{\zeta_{\ell k,i}} \sum_{j=1}^{l(i)} \frac{\zeta_{\ell k,j}}{\mathcal{V}_j} \left( \mathcal{B}_{k,j}^{11} S_j^{x2} + \mathcal{B}_{k,j}^{22} S_j^{y2} \right) , \quad \text{(B4)}$$

with $\mathcal{B}^{11}$ and $\mathcal{B}^{22}$ referring to the diagonal entries of $\widetilde{\mathbf{G}}^{\mathrm{T}}\mathbf{C}$. Subsequently, Eq. (B3) is executed as TS10

$$e^{\mu+1} = \omega [\mathcal{D} - \mathcal{P}_z]^{-1} (\mathcal{D} e^{\mu} - \mathcal{P}_{\mathrm{h}}(e^{\mu}) + \hat{r}) + (1 - \omega) e^{\mu} \quad \text{(B5)}$$

with the weight $\omega = 0.7$.

---

[17]Compared to the unsplit scheme, the particular horizontal–vertical splitting also does not incur any additional parallel communication in the context of the horizontal domain decomposition of IFS-FVM.

## Appendix C: Geospherical framework, generalized terrain-following vertical coordinate, shallow- and deep-atmosphere equations

IFS-FVM's ability to accommodate complex mesh geometries results from two aspects of its formulation: the horizontal unstructured-mesh FV discretization and generalized curvilinear coordinate mappings embedded in a geospherical framework (Prusa and Smolarkiewicz, 2003; Szmelter and Smolarkiewicz, 2010).

In the geospherical curvilinear coordinate framework of Prusa and Smolarkiewicz (2003), the vector $\boldsymbol{u} = [u, v, w]^T$ represents the physical velocity with zonal, meridional, and vertical components aligned at every point of the spherical shell with axes of a local Cartesian frame (marked with the superscript "c") tangent to the lower surface ($r = a$); here $r$ is the radial component of the vector radius, and $a$ is the radius of the sphere. Relations between the local Cartesian and the geospherical frame are therefore $dx_c = r \cos\phi \, d\lambda$, $dy_c = r \, d\phi$, and $z_c = r - a$, where $\lambda$ and $\phi$ denote longitude and latitude, respectively, in radians.

Consistent with Prusa and Smolarkiewicz (2003) (but not with their notation), we define a set of geospherical coordinates of the physical space $\mathbf{S}_p$ as $\widetilde{x} = a\lambda$, $\widetilde{y} = a\phi$, $\widetilde{z} = z_c$ ($\widetilde{x}, \widetilde{y}, \widetilde{z}$ in units of metres [TS11]). The latter are related to the curvilinear coordinates $\boldsymbol{x} = [x, y, z]^T$ of the computational space $\mathbf{S}_t$ (see Sect. 2.1.1) by the general transformation

$$(t, \boldsymbol{x}) = \left(\widetilde{t}, \mathcal{F}(\widetilde{t}, \widetilde{\boldsymbol{x}})\right) , \tag{C1}$$

where $\mathcal{F}(\widetilde{t}, \widetilde{\boldsymbol{x}})$ represents a bijective map between the physical and computational systems (Prusa and Smolarkiewicz, 2003; Kühnlein et al., 2012). A default mapping in IFS-FVM uses no stretching with respect to the horizontal positions of the unstructured computational mesh $x \equiv \widetilde{x}$, $y \equiv \widetilde{y}$, combined with a height-based terrain-following vertical coordinate of the general form $z = z(\widetilde{x}, \widetilde{y}, \widetilde{z})$. The most straightforward specification of the mapping $z = z(\widetilde{x}, \widetilde{y}, \widetilde{z})$ is a basic terrain-following vertical coordinate given by means of analytical functions; e.g. Gal-Chen and Somerville (1975). However, the implemented general form $z = z(\widetilde{x}, \widetilde{y}, \widetilde{z})$ admits variable vertical stretching with horizontal location, implicit–explicit smoothing of coordinate levels (Schär et al., 2002; Klemp, 2011), and hybrid specifications. Further note that in the numerical experiments of Sect. 3, IFS-FVM employed the vertical levels defined by the height of the hybrid sigma–pressure coordinate levels of IFS-ST. Overall, under the described coordinate mappings, the $3 \times 3$ coefficient matrix $\widetilde{\mathbf{G}}$ employed in the formalism of Sect. 2.1.1 is given as

$$\widetilde{\mathbf{G}} = \begin{bmatrix} \widetilde{G}_1^1 & \widetilde{G}_1^2 & \widetilde{G}_1^3 \\ \widetilde{G}_2^1 & \widetilde{G}_2^2 & \widetilde{G}_2^3 \\ \widetilde{G}_3^1 & \widetilde{G}_3^2 & \widetilde{G}_3^3 \end{bmatrix} =$$
$$\begin{bmatrix} \left(\Gamma\cos(\widetilde{y}/a)\right)^{-1} & 0 & \left(\Gamma\cos(\widetilde{y}/a)\right)^{-1}\partial z/\partial\widetilde{x} \\ 0 & \Gamma^{-1} & \Gamma^{-1}\partial z/\partial\widetilde{y} \\ 0 & 0 & \partial z/\partial\widetilde{z} \end{bmatrix}, \tag{C2}$$

where

$$\Gamma = 1 + \gamma \, \widetilde{z}/a , \tag{C3}$$

with $\gamma = 0$ and $\gamma = 1$ for the shallow- and deep-atmosphere form of the governing Eqs. (1a)–(1e), respectively, and the indices 1, 2, and 3 correspond to $x$, $y$, and $z$ components. The corresponding Jacobian of the coordinate mappings $\mathcal{G}$, which appears in the system (1a)–(1e)[TS12] as well as other equations, is

$$\mathcal{G} = \Gamma^2 \cos(\widetilde{y}/a)(\partial z/\partial\widetilde{z})^{-1}. \tag{C4}$$

The inverse metrics $\partial z/\partial\widetilde{x}$, $\partial z/\partial\widetilde{y}$ and $\partial z/\partial\widetilde{z}$ in Eqs. (C2) and (C4) are computed consistently with the FV discretization of Sect. 2.1.3 and using the Kronecker-delta identity; e.g. Kühnlein et al. (2012).

Furthermore, in the momentum Eq. (1b), the components of the Coriolis acceleration are

$$-\boldsymbol{f} \times \boldsymbol{u} = \big[ v \, f_0 \sin(\widetilde{y}/a) - \gamma \, w \, f_0 \cos(\widetilde{y}/a) ,$$
$$- u \, f_0 \sin(\widetilde{y}/a) , \gamma \, u \, f_0 \cos(\widetilde{y}/a) \big]^T , \tag{C5}$$

where $f_0 = 2|\boldsymbol{\Omega}|$, and the metric forcings due to the curvature of the sphere (i.e. component-wise Christoffel terms associated with the advective derivative of the physical velocity) are

$$\mathcal{M}(\boldsymbol{u}) = (\Gamma a)^{-1}\big[ \tan(\widetilde{y}/a)\, u \, v - \gamma \, u \, w , -\tan(\widetilde{y}/a)\, u \, u$$
$$- \gamma \, v \, w , \gamma \, (u \, u + v \, v) \big]^T . \tag{C6}$$

The buoyancy term of Eq. (1b) contains the gravitational acceleration $g$, which is given as

$$g = g_0 \, \Gamma^{-2} . \tag{C7}$$

Although not applied in the present paper, the optional time dependence of the generalized curvilinear coordinates enters through the mesh velocity $\boldsymbol{v}^g$, as indicated in Sect. 2.1.1; see Prusa and Smolarkiewicz (2003) and Kühnlein et al. (2012) for further discussion.

## Appendix D: Summary of variables and physical constants

**Table D1.** List of variables.

| Symbol | Description | Units |
|---|---|---|
| $\lambda$ | Longitude | rad |
| $\phi$ | Latitude | rad |
| $z$ | Height with respect to mean sea level for which it is set to zero | m |
| $r$ | Radial distance | m |
| $p_s$ | Surface hydrostatic pressure, used with the HPEs in IFS-ST | Pa |
| $\pi_s$ | Surface hydrostatic pressure, used with the fully compressible equations in IFS-ST | Pa |
| $u$ | Zonal wind velocity | $\mathrm{m\,s^{-1}}$ |
| $v$ | Meridional wind velocity | $\mathrm{m\,s^{-1}}$ |
| $w$ | Vertical wind velocity | $\mathrm{m\,s^{-1}}$ |
| $\boldsymbol{u}$ | 3-D wind vector | $\mathrm{m\,s^{-1}}$ |
| $p$ | Pressure | Pa |
| $\rho$ | Total air density | $\mathrm{kg\,m^{-3}}$ |
| $\rho_d$ | Dry air density | $\mathrm{kg\,m^{-3}}$ |
| $T$ | Temperature | K |
| $T_v$ | Virtual temperature | K |
| $\theta$ | Potential temperature | K |
| $\theta_\rho$ | Density potential temperature | K |
| $d_4$ | Vertical divergence variable, used with the fully compressible equations in IFS-ST | $\mathrm{s^{-1}}$ |
| $\hat{q}$ | Nonhydrostatic pressure departure, used with the fully compressible equations in IFS-ST | 1 |
| $\pi$ | Exner pressure | 1 |
| $\varphi$ | Normalized Exner pressure | $\mathrm{J\,kg^{-1}\,K^{-1}}$ |
| $r_k$ | Mixing ratio moisture variables (vapour $r_v$, liquid $r_l$, rain $r_r$, ice $r_i$, snow $r_s$) | $\mathrm{kg\,kg^{-1}}$ |
| $q_k$ | Specific moisture variables (vapour $q_v$, liquid $q_l$, rain $q_r$, ice $q_i$, snow $q_s$) | $\mathrm{kg\,kg^{-1}}$ |

**Table D2.** List of physical constants.

| Constant | Description | Value |
|---|---|---|
| $g_0$ | Gravitational acceleration at the Earth's surface | $9.80616\,\mathrm{m\,s^{-2}}$ |
| $a$ | Earth's mean radius | $6.371229 \times 10^6\,\mathrm{m}$ |
| $\Omega \equiv |\boldsymbol{\Omega}|$ | Earth's angular velocity | $7.29212 \times 10^{-5}\,\mathrm{m^{-1}}$ |
| $p_0$ | Reference pressure | 1000 hPa |
| $c_{pd}$ | Specific heat capacity of dry air at constant pressure | $1004.5\,\mathrm{J\,kg^{-1}\,K^{-1}}$ |
| $c_{vd}$ | Specific heat capacity of dry air at constant volume | $717.5\,\mathrm{J\,kg^{-1}\,K^{-1}}$ |
| $R_d$ | Gas constant for dry air | $287.0\,\mathrm{J\,kg^{-1}\,K^{-1}}$ |
| $R_v$ | Gas constant for water vapour | $461.5\,\mathrm{J\,kg^{-1}\,K^{-1}}$ |
| $\varepsilon$ | Ratio of $R_d$ to $R_v$ | 0.622 |

*Author contributions.* CK did most of the developments and numerical experiments presented in the paper. CK and PKS are the main developers of IFS-FVM. WD is the main developer of the Atlas library employed by IFS-FVM. RK contributed to developments of the advection scheme and time stepping. SM set IFS-ST up for the test cases and performed experiments for the comparison to IFS-FVM. ZPP contributed to the efficient coding of IFS-FVM. JS contributed to the preconditioning of the Krylov-subspace solver. NPW provided support with regard to the general IFS framework.

*Competing interests.* The authors declare that they have no conflict of interest.

*Acknowledgements.* We would like to thank two reviewers for their useful comments. Helpful discussions with the physics parameterization team at ECMWF are gratefully acknowledged. This work was supported in part by funding received from the European Research Council under the European Union's Seventh Framework Programme FP7/2012/ERC (grant agreement no. 320375), the Horizon 2020 Research and Innovation Programme ESCAPE (grant agreement no. 671627), and ESIWACE (grant agreement no. 675191). Rupert Klein thanks ECMWF for support under their fellowship programme and acknowledges partial support of his contributions by the Deutsche Forschungsgemeinschaft, grant CRC 1114 "Scaling Cascades in Complex Systems", project A02. Zbigniew Piotrowski acknowledges partial support from the "Numerical weather prediction for sustainable Europe" project, carried out within the FIRST TEAM programme of the Foundation for Polish Science co-financed by the European Union under the European Regional Development Fund.

Edited by: Paul Ullrich
Reviewed by: two anonymous referees

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

**Remarks from the language copy-editor**

**Remarks from the typesetter**