# Peer review of "FVM 1.0: A nonhydrostatic finite-volume dynamical core for the IFS"

_Geoscientific Model Development, 2018_

## Referee Comment (RC1) · Anonymous Referee #1 · 22 Nov 2018

FVM1.0: A nonhydrostatic finite-volume dynamical core formulation for IFS gmd-2018-237

By C. Kuehnlein et al.

General comments

The manuscript documents some recent advances made in the development of a new finite-volume dynamical core within the Integrated Forecasting System at ECMWF. These include the introduction of a horizontal-vertical split transport scheme, variable timestepping, coupling to the physical parameterization suite through a flexible interface, and optional subcycling of dynamical steps between physics steps. Some of these developments, in combination with recoding, have led to significant improve-

ments in computational performance. At the same time, test case results are reassuringly similar to those produced using the operational spectral dynamical core. The paper is generally well written, and provides a valuable documentation of the state of development of what is potentially a future operational dynamical core. I would be happy to recommend publication subject to minor amendments to clarify a few points.

—

Specific comments

Section 2.1.2. There seem to be several levels of iteration: equations (11a) and (11b); the weakly nonlinear (19); and the line Jacobi preconditioner (appendix B), with various corresponding lagged terms (p11 line28, p30 line 8). Could the relation of these to each other be clarified, for example with a section of pseudocode?

P7: Could the authors say something about whether the perturbation form of the equations still works for more complex flows and in the presence of significant orography? Is a more complex 'ambient' state needed?

P7: Do the metric terms (caligraphic M) blow up at the poles, and, if so, does this cause any difficulty? Check for consistent font ((5) and (C3)).

P9 Table 2: Since there is no prognostic equation for Exner pressure in (1a)-(1e), could you explain why there is an off-centring parameter for the Exner perturbation?

P16 line 15: Could a reference be given for the IFS documentation?

P19 line 16: It is a shame that the baroclinic instability was not triggered in just one hemisphere, as grid imprinting can be revealed in the quiescent hemisphere. In the present case 'no signs of any significant grid imprinting' (P19 line 31) is rather a weak statement because the fully developed wave could be hiding any grid imprinting.

—

Minor points, typos, etc

P2 line 22: advancements -> advances

P2: I think footnote 1 could be omitted; this is a different kind of 'spectral' and readers are not likely to get confused.

P8, lines 25 and 27: there is some repetition here.

P9 line 1: there two -> there are two

P10 line 22: gather -> gathering

P11 line 4: More idiomatic would be 'still requires the Exner presure perturbation to be specified'

P11 line 24: 'respectively (7)' ??

P19 line 29: 30 ??

---

## Referee Comment (RC2) · Anonymous Referee #2 · 7 Dec 2018

**Review of the GMDD Manuscript:**
FVM 1.0: A nonhydrostatic finite-volume dynamical core formulation for IFS

**Authors:** Christian Kühnlein, Willem Deconinck, Rupert Klein, Sylvie Malardel, Zbigniew P. Piotrowski, Piotr K. Smolarkiewicz, Joanna Szmelter, and Nils P. Wedi

**General comments:**
The manuscript describes the design of the new nonhydrostatic dynamical core of ECMWF's Integrated Forecasting System (IFS) and compares it against IFS's traditional primitive-equation-based semi-Lagrangian spectral transform (IFS-ST) dynamical core. The new Finite-Volume Module of IFS (IFS-FVM) can be configured as both a shallow-atmosphere and deep-atmosphere model. The paper provides an in-depth description of the temporal and spatial discretizations, the computational grid, computational efficiency, and the physics-dynamics coupling. Both IFS-FVM and IFS-ST are compared via a baroclinic instability test case that was also used for the Dynamical Core Model Intercomparison Project (DCMIP) in 2016 (with slight variations). The numerical tests are conducted in a dry and idealized-moist configuration. The latter utilizes IFS's microphysics parameterization that triggers rainfall patterns along the developing frontal zones.
The manuscript is very well written. It includes most aspects of the model design (with the exception of the generalized vertical coordinate and the inclusion of topography, and some details about the deep atmosphere configuration) and will be a highly valuable resource in the literature. It is a comprehensive and very dense overview of the new dynamical core. Are there plans to also write an extensive scientific documentation, like an ECMWF Technical Memorandum, that could provide further details about the algorithms? It is interesting to see that IFS-FVM does not utilize any explicitly-added diffusion mechanisms. The diffusive properties are controlled by the implicit diffusion that also suppresses the computational artifacts of the co-located grid staggering (at least for this test case). The IFS-FVM and IFS-ST numerical results highly mimic each other. It demonstrates that the new dynamical core is accurate and ready for further more complex evaluations. However, the authors should take the following minor suggestions and clarifications into account before the final publication can be recommended.

**Detailed comments:**

1) Page 5, Table 1: Isn't the surface pressure forecast variable in IFS-ST $\ln(p_s)$ instead of $p_s$?

2) Page 6, after Eq. (1): Point to the Tables D1 and D2 right away for the explanations of the variables and physical constants. The first reference to the tables is currently on page 7 line 28 which is too late.

3) Page 6, Eq. (1):. Have you also experimented with moist adjustments of the physical constants like $c_p$?

4) Page 6, line 25: spell out 'rhs' here (first appearance of the acronym)

5) Page 7, Eq. (4): Please provide additional explanations how the ambient state is determined. Is the balance presented here computed numerically? This paper uses a stationary atmosphere in thermal wind balance as an example, but keeps it open whether this is always the best choice or whether the ambient state can (or must) also be time-dependent for more complex flows.

6) Page 7, line 24: Refer to Appendix C for further explanations of the Coriolis term (not just for the metric term). Also point out that Appendix C discusses the differences between the shallow- and deep-atmosphere equation sets for **f** and **M**.

7) Page 8, Eq. (7): Point to the Kühnlein and Smolarkiewicz (2017) paper right away to explain the meaning of the operator $A_i$.

8) Page 11, bottom: From the description of the semi-implicit time integration it is not clear whether this algorithms avoids any global communication. Is the iterative preconditioned GCR solution technique

entirely local without global communication? Please clarify.

9) Page 13, Fig. 1: the grid lines are too faint when printed, e.g. the open circles are barely visible. Reprint the figure with thicker contours.

10) Page 13, line 5: When switching to the deep-atmosphere equation set the surface area and volume of an element are no longer independent of height. Please clarify whether this statement only refers to the shallow-atmosphere configuration.

11) Page 14, section 2.1: The biggest omission in section 2.1 is the description of the generalized vertical coordinate and an explanation how topography is included. Please add this information for completeness, despite the fact that the presented test case does not have topography. Where is the lowest model level located in FVM: directly at the ground with z=0 or at about 12 m? Do you use equidistant grid spacing in the vertical direction in FVM? What is the accuracy of the finite-difference technique in the vertical, when non-equidistant grid spacing is used, e.g. does it reduce to a first-order numerical scheme?

12) Page 16/17, section 2.3: Describe how FVM with the height-based vertical coordinate deals with the fact that IFS's physical parameterizations assume a constant-pressure framework. The physics are not allowed to change the pressure. How is the pressure adjustment (after the variations of moisture) handled? Do you couple the IFS physics to FVM in an 'anelastic' way as analyzed by Malardel (ECMWF Workshop on Non-hydrostatic Modelling, 8-10 November 2010)?

13) Page 17, line 20: Please clarify whether the octahedral grid always adds about 20 points to the first latitude close to the poles regardless of the anticipated grid spacing.

Page 18, Fig. 2: I recommend re-plotting the right (color) figure and picking a more adequate color range for the grid spacings. E.g. the range between 4-7 km is not present and the colors are not used. Just display the range 7-9.5 km with a finer spacing like 0.25 km to enhance the near the details near the poles.

Page 19, line 29: the '30' seems to be spurious. Explain in the text (not just in the caption) which variable is shown in Figs. 3 and 4.

Page 20: Fig. 3, also other figures: Explain how this comparison was conducted. Since FVM uses a height coordinate and ST a pressure-based vertical coordinate, a comparison at the lowest full model level does not display the same cross sections. Do you use vertical interpolation/extrapolations to bring the results to the same vertical positions? Or do you assume that the variations of the vertical positions are so small (how small, please quantify) that you ignore them here. Please clarify.

Page 22, line 12: remove one of the double 'to to'

Page 27: Please comment briefly on the domain decompositions for FVM and ST. Are ghost cell added to the parallel domains? If yes, how wide are they?

Page 30, Appendix C: The appendix should be renamed to also reflect the fact that it contains the discussion of the shallow- versus deep-atmosphere configuration. There seems to be an inaccuracy how the deep atmosphere equations are formulated. In line 25, the deep atmosphere should use 'r' instead of the fixed Earth's radius 'a' in the formulation of geospherical coordinates in the horizontal direction. This also requires changes of related equations on page 31 that display the radius 'a' instead of 'r'. For example, the metric term in Eq. C3 is formulated with 'a' instead of 'r' for the deep atmosphere. Please clarify this issue and correct. How is this implemented in the model FVM?

Page 32, Table D1: add the symbol 'r' to the Table. Note that 'r' is also used in a different context as a residual in Appendix B which should be changed (e.g. with a subscript).

Page 33, Table D2: add the values of the Earth's radius 'a' and the Earth's angular velocity |Omega| to the table of physical constants.

---

## Author Comment (AC1) · 17 Jan 2019

Replies Reviewer #1:

Please note that changes to the paper in response to reviewers 1 and 2 are highlighted in blue and red colours, respectively, while the colour green indicates changes that refer to both reviewers. The colour magenta marks our own revisions implemented to improve the presentation. Cosmetic changes improving grammar and style are not marked.

**1.) Section 2.1.2. There seem to be several levels of iteration: equations (11a) and (11b); the weakly nonlinear (19); and the line Jacobi preconditioner (appendix B), with various corresponding lagged terms (p11 line28, p30 line 8). Could the relation of these to each other be clarified, for example with a section of pseudocode?**

We have extended the text in Section 2.1.2 with more detailed information about the iterative approach in IFS-FVM, and in particular how the outer iterations in the model algorithm relate to the GCR solution and its preconditioner;
pg. 10 lines 17-20, pg. 11 line 28 to pg. 12 line 15, pg. 21 lines 12-16., pg. 34 line 7.

**2.) P7: Could the authors say something about whether the perturbation form of the equations still works for more complex flows and in the presence of significant orography? Is a more complex 'ambient' state needed?**

The ambient state can be as simple as vertical profiles in hydrostatic balance, which is a minimal specification that works generally, including complex flows with real orography. However, such a minimal specification is not optimal for NWP and climate modelling. In the authors' experience, customized multidimensional specifications can substantially simplify model initialization, boundary conditions and benefit the overall robustness and accuracy of the semi-implicit integration. We have extended the text on pg. 7 lines 17-19 to address this comment.

**3.) P7: Do the metric terms (caligraphic M) blow up at the poles, and, if so, does this cause any difficulty? Check for consistent font ((5) and (C3)).**

The longitude-latitude system has singularities at the poles that is well understood; cf. Prusa JCP 2018 and references therein. The octahedral mesh and all other meshes employed by IFS-FVM are specified such that there are no nodes directly at the poles. The nodes are always arranged around the poles, and the information on the polar singularity is accounted for in the finite-volume differencing across the poles; see Szmelter and Smolarkiewicz JCP 2010. Consequently, the metric terms $\mathbf{\mathcal{M}}$ are well defined and do not cause any difficulty. The font in (5) and (C3) has been made consistent.

**4.) P9 Table 2: Since there is no prognostic equation for Exner pressure in (1a)-(1e), could you explain why there is an off-centring parameter for the Exner perturbation?**

To explain, we have extended Table 2 (pg. 9), the text on pg. 11 lines 22-25 and pg. 12 lines 18-23, and added new results in Fig. 5 (middle panel). This comprehensively addresses the reviewer's comment.

**5.) P16 line 15: Could a reference be given for the IFS documentation?**

On pg. 15 lines 20-21, we refer now explicitly to Wedi et al. 2015, which is a recent overview of IFS-ST and provides a comprehensive list of references as well as the IFS documentation that is available from the ECMWF website.

***6.) P19 line 16: It is a shame that the baroclinic instability was not triggered in just one hemisphere, as grid imprinting can be revealed in the quiescent hemisphere. In the present case 'no signs of any significant grid imprinting' (P19 line 31) is rather a weak statement because the fully developed wave could be hiding any grid imprinting.***

To corroborate our statement in the paper we have now added a comparison of IFS-FVM and IFS-ST dry simulation results at day 15, where the trigger is applied only in the northern hemisphere. This is shown now in Fig. 6, the original result with the dual triggering in the northern and southern hemispheres is still given in Fig. 5. The new Fig. 6 shows that for the applied O320 grid, there is indeed no significant imprinting at day 15 visible in the depicted surface pressure field. Grid imprinting generally becomes smaller with higher resolution, and the O320 grid is the coarsest grid that will ever be used by IFS-FVM at ECMWF. Moreover, IFS-FVM is participating in the DCMIP-2016 model intercomparison, and a corresponding paper about the baroclinic instability test case will soon be submitted to GMD.

***7.) Minor points, typos, etc.***

All done. Thank you.

Replies Reviewer #2:

Please note that changes to the paper in response to reviewers 1 and 2 are highlighted in blue and red colours, respectively, while the colour green indicates changes that refer to both reviewers. The colour magenta marks our own revisions implemented to improve the presentation. Cosmetic changes improving grammar and style are not marked.

***1.) Page 5, Table 1: Isn't the surface pressure forecast variable in IFS-ST ln(ps) instead of ps?***

The reviewer is right. We have modified Table 1 accordingly.

***2.) Page 6, after Eq. (1): Point to the Tables D1 and D2 right away for the explanations of the variables and physical constants. The first reference to the tables is currently on page 7 line 28 which is too late.***

Done. The reference to the Tables D1 and D2 has been moved right after Eq. (1).

***3.) Page 6, Eq. (1):. Have you also experimented with moist adjustments of the physical constants like $c_p$?***

Due to the excellent agreement of the IFS-FVM results with the IFS-ST under moist/precipitating dynamics with the coupling to the IFS cloud scheme, we have not considered this to be a priority. However, the impact of moisture on thermodynamic parameters will be addressed in the near future.

***4.) Page 6, line 25: spell out 'rhs' here (first appearance of the acronym).***

Done.

***5.) Page 7, Eq. (4): Please provide additional explanations how the ambient state is determined. Is the balance presented here computed numerically? This paper uses a stationary atmosphere in thermal wind balance as an example, but keeps it open whether this is always the best choice or whether the ambient state can (or must) also be time-dependent for more complex flows.***

In the baroclinic instability test case, the ambient state is specified using the analytic initial condition of the test case minus the triggering perturbation. This state is in thermal wind balance, while being variable in latitude and in height, but invariant in longitude; see pg. 20 lines 26-29. The optimal specification of ambient states for NWP and climate modelling is ongoing research. In general, ambient states can be as simple as stationary vertical profiles in hydrostatic balance, but in the authors' experience customized specifications for a particular application can benefit the accuracy and robustness of the model integration; see the text on pg. 7 lines 17-19.

***6.) Page 7, line 24: Refer to Appendix C for further explanations of the Coriolis term (not just for the metric term). Also point out that Appendix C discusses the differences between the shallow- and deep atmosphere equation sets for f and M.***

Done; pg. 7 lines 31-32.

**7.) Page 8, Eq. (7): Point to the Kühnlein and Smolarkiewicz (2017) paper right away to explain the meaning of the operator $A_i$.**

Done; pg. 8 lines 18-19.

**8.) Page 11, bottom: From the description of the semi-implicit time integration it is not clear whether this algorithms avoids any global communication. Is the iterative preconditioned GCR solution technique entirely local without global communication? Please clarify.**

The GCR solver uses global communication for the computation of inner products and residual error norms. We have extended the text accordingly on pg. 15 lines 5-9.

**9.) Page 13, Fig. 1: the grid lines are too faint when printed, e.g. the open circles are barely visible. Reprint the figure with thicker contours.**

Done.

**10.) Page 13, line 5: When switching to the deep-atmosphere equation set the surface area and volume of an element are no longer independent of height. Please clarify whether this statement only refers to the shallow-atmosphere configuration.**

The presented formulation applies to both the shallow- and deep-atmosphere equations. The surface area and volume of the dual cells, as well as delta z, are quantities of the computational space, and these are independent of height. Deep-atmosphere effects such as the widening of the vertical columns with height is present in physical space, which is accounted for by the curvilinear coordinate framework described in Appendix C. We have extended Section 2.1.3 including an example of the velocity divergence in physical space, pg. 13 lines 13 and 15, pg. 14 lines 6-12. This exposes the connection of the operators in computational space with the corresponding physical-space quantities.

**11.) Page 14, section 2.1: The biggest omission in section 2.1 is the description of the generalized vertical coordinate and an explanation how topography is included. Please add this information for completeness, despite the fact that the presented test case does not have topography. Where is the lowest model level located in FVM: directly at the ground with z=0 or at about 12 m? Do you use equidistant grid spacing in the vertical direction in FVM? What is the accuracy of the finite-difference technique in the vertical, when non-equidistant grid spacing is used, e.g. does it reduce to a first-order numerical scheme?**

We have added the description of the generalized terrain-following vertical coordinate and its specification at the beginning of Section 3 (pg. 20 lines 4-15) and Appendix C (pg. 35 lines 4-8). Our design based on curvilinear coordinates warrants second-order accuracy for non-equidistant grid spacing in the physical space (see Kühnlein et al. JCP 2012 and references therein).

**12.) Page 16/17, section 2.3: Describe how FVM with the height-based vertical coordinate deals with the fact that IFS's physical parameterizations assume a constant-pressure framework. The physics are not**

*allowed to change the pressure. How is the pressure adjustment (after the variations of moisture) handled?*

*Do you couple the IFS physics to FVM in an 'anelastic' way as analyzed by Malardel (ECMWF Workshop on Non-hydrostatic Modelling, 8-10 November 2010)?*

Correct, the IFS physical parametrisations assume constant pressure. This applies also when called from IFS-FVM. How the pressure adjustment is handled is already described in the paper; see the discussion around Eq. (17). It represents a 'compressible' (in contrast to 'anelastic') coupling, when referring to Malardel, ECMWF Workshop of Non-hydrostatic Modelling, 8-10 November 2010. However, in contrast to the internal energy form, in IFS-FVM the thermodynamic equation (or first law of thermodynamics) is for potential temperature and thus in enthalphy form. We have specified the forcing from the parametrisations to the prognostic pressure equation (17) in the new Eq. (19), and adjuted the text accordingly (pg. 11 lines 21-22).

**13.) Page 17, line 20: Please clarify whether the octahedral grid always adds about 20 points to the first latitude close to the poles regardless of the anticipated grid spacing.**

The standard procedure is 20 points on the first latitude, at least for mesh sizes between O90 (120km) and O8000 (1km). However, somewhat larger values between 24 and 28 could be used in the future. We revised the text on pg. 18 line 17.

**14.) Page 18, Fig. 2: I recommend re-plotting the right (color) figure and picking a more adequate color range for the grid spacings. E.g. the range between 4-7 km is not present and the colors are not used. Just display the range 7-9.5 km with a finer spacing like 0.25 km to enhance the near the details near the poles.**

We changed the range of the colour bar from 7-9.5 km, but kept the spacing at 0.5 km, as 0.25 km does not really reveal more detail near the poles.

**15.) Page 19, line 29: the '30' seems to be spurious. Explain in the text (not just in the caption) which variable is shown in Figs. 3 and 4.**

Yes, the number '30' must have entered by accident. We have added the shown variables to the main text on pg. 21 lines 4-5.

**16.) Page 20: Fig. 3, also other figures: Explain how this comparison was conducted. Since FVM uses a height coordinate and ST a pressure-based vertical coordinate, a comparison at the lowest full model level does not display the same cross sections. Do you use vertical interpolation/extrapolations to bring the results to the same vertical positions? Or do you assume that the variations of the vertical positions are so small (how small, please quantify) that you ignore them here. Please clarify.**

Done. It is explained in the introduction to Section 3 on pg. 20 lines 4-15.

**17.) Page 22, line 12: remove one of the double 'to to'**

Done.

**18.) Page 27: Please comment briefly on the domain decompositions for FVM and ST. Are ghost cell added to the parallel domains? If yes, how wide are they?**

We have extended the existing text about the domain decomposition with more details such as halo width; pg. 15 lines 5-8.

**19.) Page 30, Appendix C: The appendix should be renamed to also reflect the fact that it contains the discussion of the shallow- versus deep-atmosphere configuration. There seems to be an inaccuracy how the deep atmosphere equations are formulated. In line 25, the deep atmosphere should use 'r' instead of the fixed Earth's radius 'a' in the formulation of geospherical coordinates in the horizontal direction. This also requires changes of related equations on page 31 that display the radius 'a' instead of 'r'. For example, the metric term in Eq. C3 is formulated with 'a' instead of 'r' for the deep atmosphere. Please clarify this issue and correct. How is this implemented in the model FVM?**

We have again verified what was written. All provided formulae with respect to 'r'and 'a' are correct for both the shallow- and the deep-atmosphere equations. However, for clarity, we have revised Appendix C, and now have a separate numbered equation for $\Gamma$ in (C3) that exposes the switch between deep- and shallow-atmosphere equations.

**20.) Page 32, Table D1: add the symbol 'r' to the Table. Note that 'r' is also used in a different context as a residual in Appendix B which should be changed (e.g. with a subscript).**

Done. We use now $\hat{r}$ for the residual error when discussing the Line Jacobi preconditioner in Appendix B.

**21.) Page 33, Table D2: add the values of the Earth's radius 'a' and the Earth's angular velocity |Omega| to the table of physical constants.**

Done.
* * *
Many thanks.

[revised manuscript text omitted]